# Activation of ILC2s through constitutive IFNγ signaling reduction leads to spontaneous pulmonary fibrosis

Natsuko Otaki[1,2], Yasutaka Motomura[1,3,4], Tommy Terooatea[5], S. Thomas Kelly[5], Miho Mochizuki[6], Natsuki Takeno[6], Shigeo Koyasu [2,6], Miu Tamamitsu[7], Fuminori Sugihara[8], Junichi Kikuta [9], Hideya Kitamura[10], Yoshiki Shiraishi [11], Jun Miyanohara[12], Yuji Nagano [12], Yuji Saita[12], Takashi Ogura[10], Koichiro Asano[11], Aki Minoda[5,13] & Kazuyo Moro [1,3,4,14] ✉

Pulmonary fibrosis (PF), a condition characterized by inflammation and collagen deposition in the alveolar interstitium, causes dyspnea and fatal outcomes. Although the bleomycin-induced PF mouse model has improved our understanding of exogenous factor-induced fibrosis, the mechanism governing endogenous factor-induced fibrosis remains unknown. Here, we find that *Ifngr1-/-Rag2-/-* mice, which lack the critical suppression factor for group 2 innate lymphoid cells (ILC2), develop PF spontaneously. The onset phase of fibrosis includes ILC2 subpopulations with a high *Il1rl1* (IL-33 receptor) expression, and fibrosis does not develop in ILC-deficient or IL-33-deficient mice. Although ILC2s are normally localized near bronchioles and blood vessels, ILC2s are increased in fibrotic areas along with IL-33 positive fibroblasts during fibrosis. Co-culture analysis shows that activated-ILC2s directly induce collagen production from fibroblasts. Furthermore, increased *IL1RL1* and decreased *IFNGR1* expressions are confirmed in ILC2s from individuals with idiopathic PF, highlighting the applicability of *Ifngr1-/-Rag2-/-* mice as a mouse model for fibrosis research.

Innate lymphoid cells (ILC) are activated not by foreign antigens, but endogenous factors such as cytokines, lipids, and neuropeptides[1,2]. Since many reports have demonstrated the importance of ILC2s in allergic diseases[3], we previously studied the suppression mechanisms of ILC2s in controlling allergic diseases and found that ILC2 function is strongly inhibited by IFNγ and IL-27[4]. Based on this finding, we established *Ifngr1-/-Rag2-/-* mice to analyze the function of IFNγ in ILC2s without the acquired immune system, which was achieved by deleting

[1]Laboratory for Innate Immune Systems, RIKEN Center for Integrative Medical Sciences (IMS), Kanagawa, Japan. [2]Department of Microbiology and Immunology, Keio University School of Medicine, Tokyo, Japan. [3]Laboratory for Innate Immune Systems, Department of Microbiology and Immunology, Graduate School of Medicine, Osaka University, Osaka, Japan. [4]Laboratory for Innate Immune Systems, Immunology Frontier Research Center (IFReC), Osaka University, Osaka, Japan. [5]Laboratory for Cellular Epigenomics, RIKEN Center for Integrative Medical Sciences (IMS), Kanagawa, Japan. [6]Laboratory for Immune Cell Systems, RIKEN Center for Integrative Medical Sciences (IMS), Kanagawa, Japan. [7]Research Center for Advanced Science and Technology, The University of Tokyo, Tokyo, Japan. [8]Central Instrumentation Laboratory, Research Institute for Microbial Diseases, Osaka University, Osaka, Japan. [9]Department of Immunology and Cell Biology, Graduate School of Medicine and Frontier Biosciences, Osaka University, Osaka, Japan. [10]Kanagawa Cardiovascular and Respiratory Center, Kanagawa, Japan. [11]Division of Pulmonary Medicine, Department of Medicine, Tokai University School of Medicine, Kanagawa, Japan. [12]Discovery Accelerator, Astellas Pharma Inc., Ibaraki, Japan. [13]Department of Cell Biology, Faculty of Science, Radboud Institute for Molecular Life Sciences, Radboud University, Nijmegen, Netherlands. [14]Laboratory for Innate Immune Systems, Graduate School of Frontier Biosciences, Osaka University, Osaka, Japan. ✉e-mail: moro@ilc.med.osaka-u.ac.jp

the *Rag2* gene. Using young *Ifngr1*[-/-]*Rag2*[-/-] mice, we reported that the loss of IFNγ signaling exacerbates allergic reactions[4]. Importantly, we also noticed that pulmonary fibrosis (PF) develops spontaneously in aged *Ifngr1*[-/-]*Rag2*[-/-] mice. Since 20% of *Rag2*[-/-] mice are known to develop fibrosis spontaneously with aging[5], we were not surprised by the onset of fibrosis in mice of the *Rag2*[-/-] background. However, we were interested in the contribution of IFNγ in the development of fibrosis in *Ifngr1*[-/-]*Rag2*[-/-] mice, as fibrosis was more prevalent in *Ifngr1*[-/-]*Rag2*[-/-] mice, than that reported in *Rag2*[-/-] mice[5].

Clinically, interstitial lung diseases (ILD) are classified into two main categories: idiopathic interstitial pneumonias (IIP) and secondary ILDs. IIPs can be further subdivided into nine diseases[6]. PF is a specific type of ILD characterized by extracellular matrix deposition (such as collagen) in the alveolar interstitium, causing dyspnea[7,8]. Among the IIPs, idiopathic pulmonary fibrosis (IPF), a progressive disease of unknown cause, has poor prognosis, with no effective treatment[7,8]. Although the pathomechanism of IPF is not yet fully elucidated, tissue repair in response to repeated inflammation is thought to be involved in its development[7]. Exogenous factors such as smoking and dust exposure: and endogenous factors such as aging, inflammation, and mechanical stress, have been identified as factors that increase the risk of developing IPF[7]. Understanding the interplay between these factors is crucial in comprehending the etiology of IPF. Notably, the incidence of IPF rises significantly with age, thereby highlighting the association between aging and disease development[7,9]. To understand the irreversible progression of IPF with age, it is crucial to analyze the pathology separately at each progression level. However, analyzing the pathology of IPF over time in human patients is difficult due to the risks associated with biopsies, which can cause irritation and worsen the disease[10].

Several PF mouse models have been developed to date, including the bleomycin-induced and TGF-β-overexpressing models[11]. Although the bleomycin-induced model has contributed significantly to our understanding of PF by elucidating the involvement of epithelial cell damage and macrophage activation[12], it is difficult to investigate the mechanisms underlying the irreversible progression of PF using this model, because of the major limitation that the fibrosis is transient in this model and will eventually be healed[13]. Furthermore, the exogenous administration of bleomycin in this model hinders the analysis of endogenous factors involved in the development of the disease. On the other hand, studies using mice overexpressing endogenous factors, such as TGF-β and IL-13, have contributed to our understanding of the mechanism underlying endogenous factor-mediated fibrosis. Although these overexpression models are useful for analyzing phenomena that occur downstream of the factors, it is difficult to explore the causes of fibrosis that exist upstream of the factors. Additionally, it is important to consider the deviation from physiological expression levels. Consequently, there is a need for a mouse model where PF develops spontaneously and irreversibly with aging, thereby allowing for a more comprehensive understanding of its underlying mechanisms. In this respect, the *Ifngr1*[-/-]*Rag2*[-/-] mice we have established in this study show promise as a PF mouse model, as they exhibit spontaneous development of PF during the aging process.

Given that IPF is not driven by antigen-specific mechanisms and that endogenous factors play a significant role in its pathogenesis, there is a possibility that ILC2s, which are activated by endogenous factors and produce tissue repair factors such as IL-4, IL-13, and amphiregulin[1], could have a more direct involvement in fibrosis, as compared to adaptive immune cells. While elevated numbers of ILC2s have been reported in the alveolar lavage fluid from individuals with IPF[14], the precise role of ILC2s in fibrosis is not yet fully understood.

In the present study, we show that *Ifngr1*[-/-]*Rag2*[-/-] mice provide a perspective that the attenuation of ILC2 inhibitory factors causes PF via gradual activation of ILC2s. Taking advantage of the spontaneous progression of PF in these mice, we divide disease progression into three phases: intact, inflammatory, and fibrotic phases. We then analyze the pathology at each phase, which was difficult to achieve in previous models. Our phase-by-phase analysis shows that the phenotypic changes of fibroblasts begin from the inflammatory phase, and clinical parameters, such as surfactant protein D (SP-D) and static lung compliance, are undetectable until the fibrotic phase. Although steroid therapy is ineffective in individuals with IPF[8], we find that fibrosis can be prevented in *Ifngr1*[-/-]*Rag2*[-/-] mice if treatment is initiated during the inflammatory phase. In addition, we clarify that the lack of IFNγ signaling results in the over-activation of ILC2s in the inflammatory phase and accelerates collagen production from fibroblasts. These results are consistent with those observed in the human IPF patient samples that we analyze, thereby underscoring their utility in elucidating the pathogenesis of IPF.

## Results

### *Ifngr1*[-/-]*Rag2*[-/-] mice spontaneously develop PF

We noticed that the lungs of *Ifngr1*[-/-]*Rag2*[-/-] mice, which lack the acquired immune system and IFNγ signaling, turned white with age (Fig. 1a). Masson's trichrome (MT) staining and micro-computed tomography (micro-CT) imaging suggested pulmonary fibrosis development in these mice (Fig. 1b and Supplementary Fig. 1a). The lungs of young *Ifngr1*[-/-]*Rag2*[-/-] mice showed normal structures but displayed cellular infiltration with fibrin accumulation at 15 weeks of age and obvious fibrosis was observed after 20 weeks of age (Fig. 1b). Based on these observations, we decided to evaluate the degree of inflammation and fibrosis in three distinct age-groups: <10-, 10–20-, and >20-week-old. To quantify the degree of inflammation, the number of bronchoalveolar lavage fluid (BALF) cells that migrate into the alveoli due to the inflammation was examined in a total of 119 (60 females and 59 males) *Ifngr1*[-/-]*Rag2*[-/-] mice (Supplementary Fig. 1b). The number of BALF cells was significantly higher in the 10–20-week-old group, as compared to that in younger group (Fig. 1c, left), and was higher in males than in females in the >20-week-old group (Fig. 1c, right). To accurately quantify fibrosis progression, collagen fibers were stained blue using MT staining. The MT-positive areas of thickened alveolar walls in the images were extracted, and their ratio to the total lung area was calculated as the fibrosis score (Supplementary Fig. 2a, b). Fibrosis scores were calculated for 27 female and 21 male mice of different ages (Supplementary Fig. 2c) and were found to increase significantly after the age of 10 weeks, as compared to that of in the younger mice (Fig. 1d, left). In addition, these scores were higher in males than in females after 20 weeks of age (Fig. 1d, right). Consistently, the collagen amount in the lungs were significantly higher in the lesions of 22-week-old *Ifngr1*[-/-]*Rag2*[-/-] mice, as compared to that in the normal tissue of 7-week-old mice (Supplementary Fig. 2d). Considering the fibrosis scores in each age-group (Fig. 1e) and BALF cell infiltration in the alveoli (Fig. 1c), inflammation appears to begin between 10 and 20 weeks of age, and fibrosis appears to more pronounced after 20 weeks of age.

Although spontaneous PF has been reported to develop in aged *Rag2*[-/-] mice[5], gross fibrosis was not observed in the wild-type (WT) mice, or *Rag2*[-/-] or *Ifngr1*[-/-] single-knockout mice at 22 weeks of age (Fig. 1f). The fibrosis scores of the *Rag2*[-/-] mice tended to increase beyond 20 weeks of age but were not significant at the age we focused on in this study (Supplementary Fig. 3a and b). Consistently, the amount of collagen in the lungs was significantly lower in the WT mice, as well as *Rag2*[-/-] or *Ifngr1*[-/-] single-knockout mice, at over 20 weeks of age, as compared to those in the lesions observed in 22-week-old *Ifngr1*[-/-]*Rag2*[-/-] mice (Supplementary Fig. 2d). To investigate the importance of IFNγ signaling in driving fibrosis, we generated IFNγ ligand and Rag-2 double-knockout mice and confirmed that PF develops in either the IFNγ receptor or ligand defect (Supplementary Fig. 3c). Based on this age-classified analysis, we defined the following disease stages to describe the progression of PF in *Ifngr1*[-/-]*Rag2*[-/-] mice:

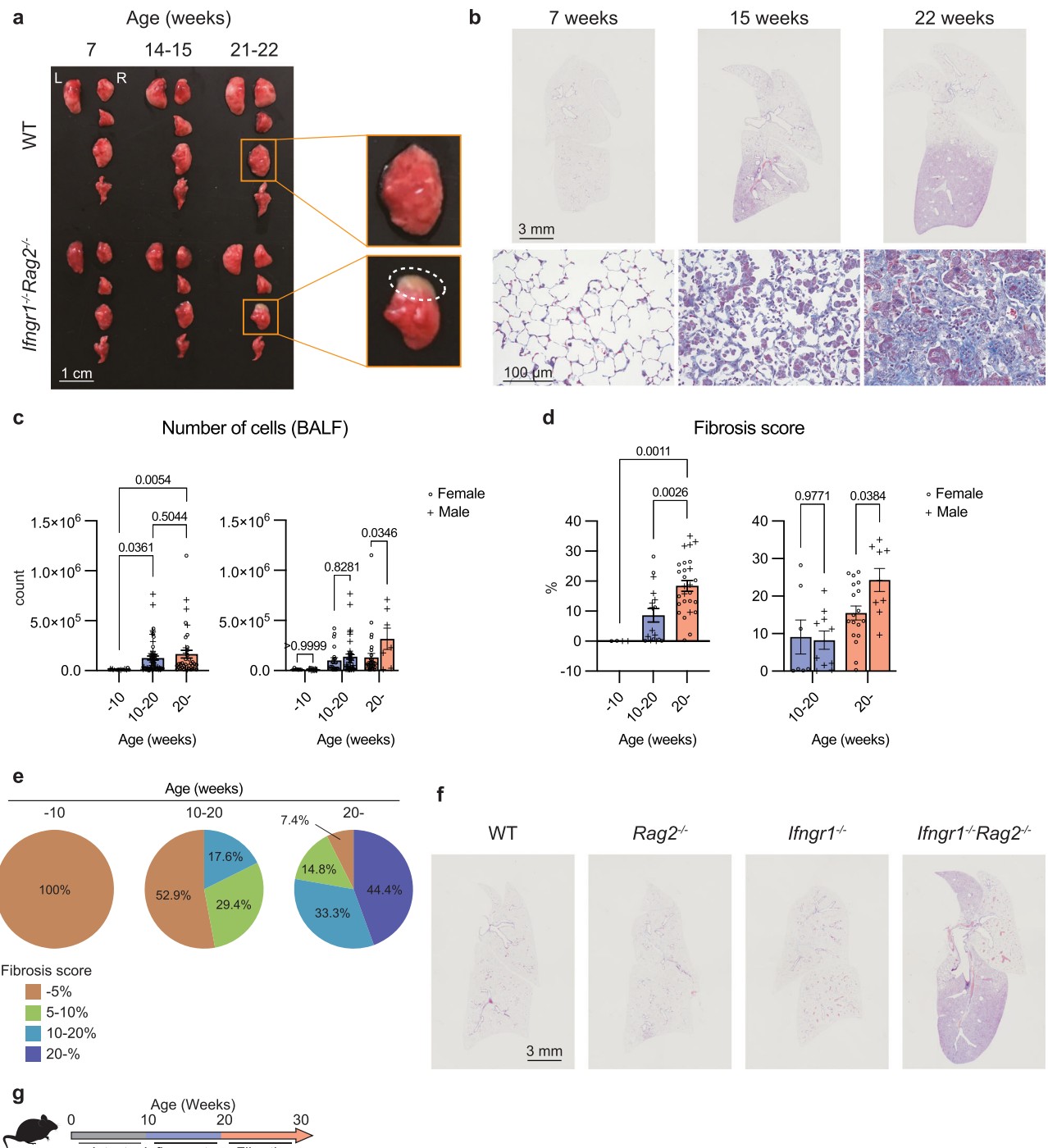

**Fig. 1 | *Ifngr1⁻/⁻Rag2⁻/⁻* mice spontaneously develop PF. a** Lungs of wild-type (WT) and *Ifngr1⁻/⁻Rag2⁻/⁻* mice. L, left; R, right. The third right lobes are magnified. A disease area, defined as a visibly distinct white region, was indicated by a dotted circle. Scale bar: 1 cm. **b** Representative Masson's trichrome (MT)-stained images of lung tissue sections from *Ifngr1⁻/⁻Rag2⁻/⁻* mice of the indicated ages (females). The upper images show the whole right lobes and the lower panels show magnified images. Scale bars: 3 mm (upper panels) or 100 μm (lower panels). **c** Quantification of the absolute number of whole bronchoalveolar lavage fluid (BALF) cells of *Ifngr1⁻/⁻Rag2⁻/⁻* mice of different ages using flow cytometry (n = 119 [60 females, 59 males]). The left graph includes data for all mice grouped by age, while the right graph shows data grouped by age and sex. See Supplementary Fig 1b for more information on each mouse. The results of several experiments were combined. **d** Fibrosis scores of *Ifngr1⁻/⁻Rag2⁻/⁻* mice calculated using MT-

stained images of their lung tissue sections (n = 48 [27 females, 21 males]). See Supplementary Fig. 2a and the Methods section for the definition of the fibrosis score. See Supplementary Fig. 2c for more information on each mouse. The results of several experiments were combined. **e** The frequency of mice with the indicated fibrosis scores, shown in Fig. 2d (n = 48 [27 females, 21 males]). The frequency was calculated for each age group. **f** Representative MT-stained images of lung tissue sections from the indicated mouse strains, showing whole right lobes (21 weeks; males). Scale bar: 3 mm. **g** Schematic of the stages of disease progression. Data are representative of at least three independent experiments and are presented as the mean ± s.e.m. For statistical analysis, the following tests were used: (**c**, **d**) (left), one-way ANOVA with Tukey's multiple comparisons tests; (**c**, **d**) (right), two-way ANOVA with Sidak's multiple comparisons tests. For (**c**, **d**) source data are provided as a Source Data file.

(i) intact phase (<10 weeks old), (ii) inflammatory phase (10–20-week-old, characterized by inflammatory cell infiltration in the BALF and mild to moderate collagen deposition in the lungs), and (iii) fibrotic phase (>20 weeks old, characterized by severe collagen deposition in the lungs) (Fig. 1g). We expected that a detailed comparison of the inflammatory and fibrotic phases of *Ifngr1⁻/⁻Rag2⁻/⁻* mice, in which fibrosis develops without external stimuli, would elucidate the mechanism of fibrosis caused by endogenous factors.

## Assessment of the pathophysiology of *Ifngr1⁻/⁻Rag2⁻/⁻* mice from a clinical perspective

We next examined the clinical characteristics that *Ifngr1⁻/⁻Rag2⁻/⁻* mice share with human PF. Honeycombing of the lungs, which is an important histological feature for diagnosing IPF in the clinic[15], was not seen in *Ifngr1⁻/⁻Rag2⁻/⁻* mice (Fig. 1b). Furthermore, the accumulation of fibrin in the alveoli was pronounced in these mice (Fig. 1b), suggesting that fibrin processing has not kept pace with the rapid disease progression. Levels of surfactant protein D (SP-D), a common clinical PF biomarker reflecting disease activity[8], increased significantly in the sera of the 30–31-week-old *Ifngr1⁻/⁻Rag2⁻/⁻* mice, as compared to that obtained from mice in the inflammatory phase (Fig. 2a). Consistently, the saturation of percutaneous oxygen (SpO₂) levels of the 30–31-week-old *Ifngr1⁻/⁻Rag2⁻/⁻* mice significantly decreased, as compared to those of the mice in the inflammatory phase (Fig. 2b). SpO₂ levels of the WT and *Rag2⁻/⁻* mice did not change with aging (Supplementary Fig. 3d). Static compliance ($C_{st}$), a factor calculated from the tidal volume, plateau pressure, and positive end-expiratory pressure (PEEP), is an indicator of lung elasticity[16]. The $C_{st}$ of 33-week-old *Ifngr1⁻/⁻Rag2⁻/⁻* mice decreased significantly as compared to that of the 14-week-old mice (Fig. 2c). These data collectively indicated that, except for some histological differences, the clinical indicators of lung function in *Ifngr1⁻/⁻Rag2⁻/⁻* mice are similar to those in human patients with PF.

Among the nine categories of IIPs, IPF is refractory, because fibrosis is irreversible and resistant to corticosteroid treatment[8]. To address whether corticosteroid treatment is effective against PF in *Ifngr1⁻/⁻Rag2⁻/⁻* mice, corticosteroids were continuously administered for 4 weeks in the inflammatory (condition 1) or fibrotic (condition 2) phase of *Ifngr1⁻/⁻Rag2⁻/⁻* mice (Fig. 2d). BALF cells were evaluated first because the number of BALF cells is low in healthy lungs but increases with fibrosis, as shown in Fig. 1c. The number of inflammatory cells in the BALF reduced after corticosteroid treatment in the condition 1 group, whereas there was no detectable effect in the condition 2 group (Fig. 2e). Importantly, as in human IPF, corticosteroids had no effect on the progression of fibrosis in the condition 2 group, whereas they completely stopped fibrosis development in the condition 1 group (Fig. 2f).

## ILC2s and ST2⁻KLRG1⁻ cells increase with PF progression

The results of the corticosteroid experiments suggested that clarifying the cellular dynamics during the inflammatory phase is important for understanding fibrosis development. To elucidate the comprehensive transcriptomic dynamics during spontaneous PF development, lung cells from the intact, inflammatory, and fibrotic phases in *Ifngr1⁻/⁻Rag2⁻/⁻* mice were analyzed by means of single-cell RNA-sequencing (scRNA-seq), with *Rag2⁻/⁻* mice used as the control (Supplementary Fig. 4a). Unsupervised clustering of the combined datasets and subsequent identification using known markers determined 24 cell-types (Supplementary Fig. 4b and 4c), and data are broken down by the three progression phases (Fig. 3a).

First, we analyzed fibroblasts, which are the main collagen source during fibrosis[17]. Fibroblasts were defined by the expression of *Col1a1* and *Pdgfra* (Fig. 3b), and divided into two groups (fibroblast 1 and fibroblast 2). The frequency of fibroblast 1 decreased with disease progression, while that of fibroblast 2 increased (Fig. 3c), suggesting

that fibroblast 2 may be involved in disease progression. Differential expression analysis between fibroblast 1 and 2 populations demonstrated that fibroblast 2 has pathogenic traits reflected by increased expression of fibrosis factors, such as *Col1a1*, *Col1a2*, *Col3a1*, *Timp1*, and *Dcn*, as compared to that in fibroblast 1 (Fig. 3d).

Next, we focused on *Ptprc*-positive immune cells (Fig. 3e) to gain insights into which immune cells may be playing a role in PF development. To visualize cells with a significantly increased frequency during the inflammatory and the fibrotic phases, as compared to the intact phase, we determined which cell-type among the *Ptprc*-positive cells changes with disease progression (Fig. 3f). We found that ILCs, basophils, dendritic cells, and macrophages increased in the inflammatory and fibrotic phases as the disease progressed. Since macrophages and ILCs have been reported to be involved in human IPF[18], we subsequently performed flow cytometric analysis of samples from the lungs of *Ifngr1⁻/⁻Rag2⁻/⁻* mice for analyzing the relationship between these cells and disease progression. We found that among macrophages (defined as CD45⁺Gr-1⁻F4/80⁺ cells; Supplementary Fig. 5a, b), the number of Siglec-F⁺CD11c⁺ macrophages decreased in both the inflammatory and fibrotic phases, while that of Siglec-F^low CD11c⁺ macrophages increased (Fig. 3g). Although interstitial macrophages are well known as CD11c low cells, a subset that express high levels of CD11c has been reported[19,20]. Therefore, based on the expression of *Adgre1* (F4/80), *Siglecf*, and *Itgax* (CD11c) in the scRNA-seq data (Supplementary Fig. 5c), Siglec-F⁺CD11c⁺ macrophages and Siglec-F^low CD11c⁺ macrophages were defined as alveolar and interstitial macrophages, respectively. On the other hand, the CD45⁺lineage⁻Thy-1⁺ population, which primarily consists of ILCs, was divided into two distinct subpopulations based on the expression of ST2 and KLRG1: ILC2s (ST2⁺KLRG1⁺ population), and ST2⁻KLRG1⁻ cells (Supplementary Fig. 5d, e). We found that the absolute number of ILC2s increased significantly in the inflammatory phase, compared to that in the intact phase (Fig. 3h), while the number of ST2⁻KLRG1⁻ cells increased significantly in both the inflammatory and fibrotic phases (Fig. 3i). These results collectively suggested that interstitial macrophages, ILC2s, and ST2⁻KLRG1⁻ cells may be involved in the pathogenesis of PF.

Histological analysis revealed that the distribution of ILC2s shows interesting changes with the onset of fibrosis. ILC2s are normally found near bronchioles and blood vessels called adventitial cuffs[21] and known to respond to IL-33 derived from epithelial cells during helminth infection. Comparison of the distribution of ILC2s in normal and pathological areas of 24-week-old *Ifngr1⁻/⁻Rag2⁻/⁻* mice with fibrosis demonstrated that GATA3⁺ ILC2s are localized near adventitial cuffs in normal areas, whereas ILC2s are scattered throughout the fibrotic areas (Fig. 3j). Additionally, the number of ILC2s was significantly increased in fibrotic areas compared to normal areas (Supplementary Fig. 5f). The marked proliferation of ILC2s in fibrotic areas suggested a strong link between ILC2s and fibrosis.

## ILCs are indispensable for PF progression

To investigate the importance of ILCs in PF, we next depleted ILCs from *Ifngr1⁻/⁻Rag2⁻/⁻* mice. Since IL2Rγ signaling is essential for the development of all ILC subsets, we generated *Ifngr1⁻/⁻Il2rg⁻/⁻Rag2⁻/⁻* mice, which are congenitally deficient in ILCs. We found that fibrosis did not occur at all in *Ifngr1⁻/⁻Il2rg⁻/⁻Rag2⁻/⁻* mice (Fig. 4a, b). To confirm the importance of IFNγR-mediated signaling in ILCs in the pathogenesis of PF, we performed bone marrow transfer. Bone marrows from *Ifngr1⁻/⁻Rag2⁻/⁻* mice or *Ifngr1⁻/⁻Il2rg⁻/⁻Rag2⁻/⁻* mice were transferred into ILCs-deficient *Il2rg⁻/⁻Rag2⁻/⁻* mice and assessed at 21–31 weeks after bone marrow transfer (Supplementary Fig. 6). Fibrosis was not detected in *Ifngr1⁻/⁻Il2rg⁻/⁻Rag2⁻/⁻* bone marrow-transferred mice, which displayed lower fibrosis scores, as compared to those in *Ifngr1⁻/⁻Rag2⁻/⁻* bone marrow-transferred mice (Fig. 4c, d). We also confirmed that the absolute number of ILC2s and ST2⁻KLRG1⁻ cells decreased significantly

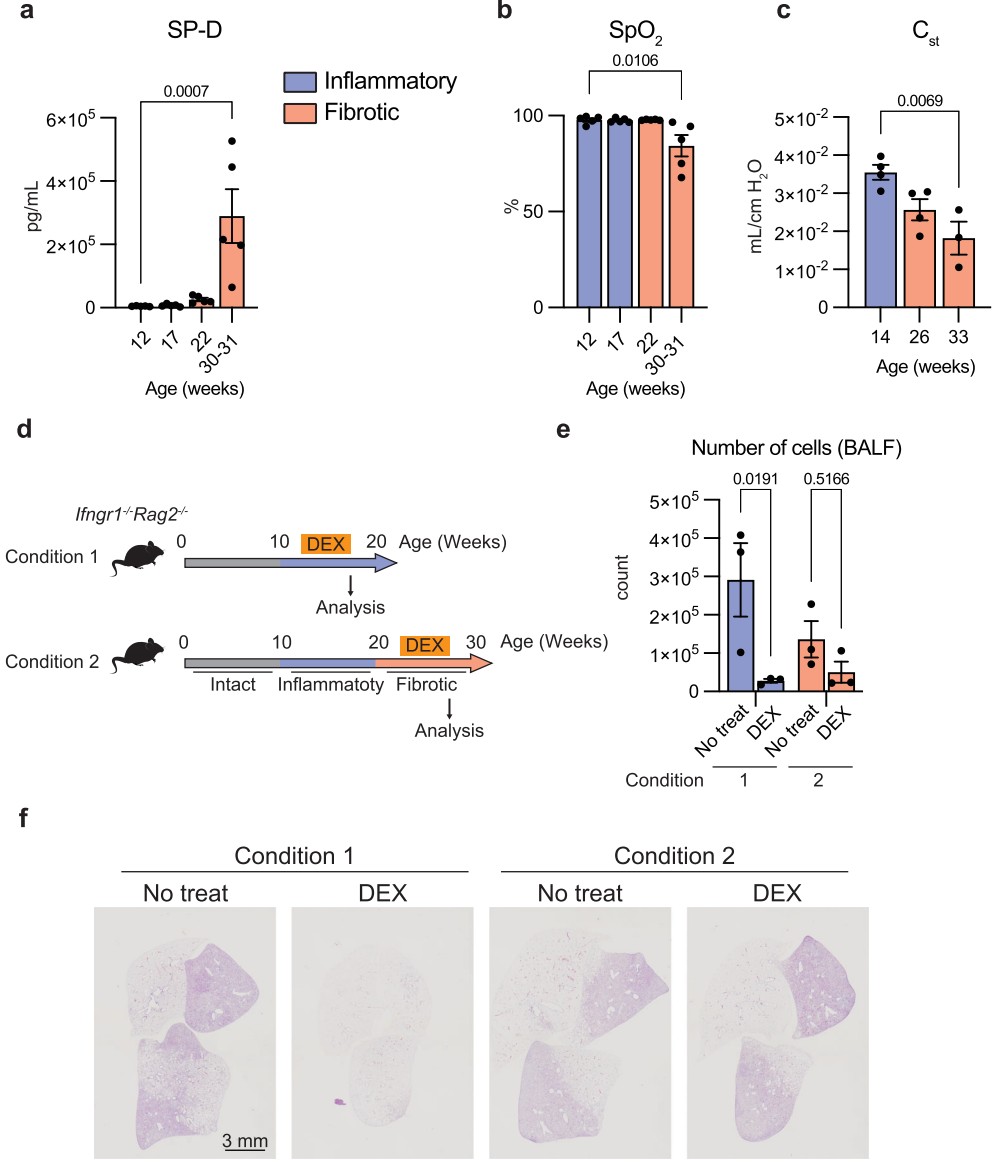

**Fig. 2 | Assessment of the pathophysiology of *Ifngr1⁻/⁻Rag2⁻/⁻* mice from a clinical perspective. a** Quantification of SP-D levels in the serum of *Ifngr1⁻/⁻Rag2⁻/⁻* mice of different ages as indicated in the graph using ELISA ($n = 5$/group; females). **b** $SpO_2$ of *Ifngr1⁻/⁻Rag2⁻/⁻* mice of different ages as indicated in the graph ($n = 5$/group; females). **c** Quantification of the $C_{st}$ of *Ifngr1⁻/⁻Rag2⁻/⁻* mice of different ages (14 and 26 weeks: $n = 4$/group; females, 33 weeks: $n = 3$/group; females). See the Methods section for the formula and the parameter of $C_{st}$. **d**–**f** Dexamethasone treatment of *Ifngr1⁻/⁻Rag2⁻/⁻* mice. DEX, dexamethasone. **d** Schematic of the experiment. *Ifngr1⁻/⁻Rag2⁻/⁻* mice were administered dexamethasone (3 mg·kg⁻¹/day) with a micro-osmotic pump implanted in the subcutaneous pockets under the back skin of mice either from 14 to 18 weeks (inflammation phase) or from 23 to 28 weeks old (fibrosis phase) ($n = 3$/group; females). **e** Quantification of the absolute number of whole BALF cells of *Ifngr1⁻/⁻Rag2⁻/⁻* mice by flow cytometry ($n = 3$/group; females). **f** Representative MT-stained images of lung tissue sections from *Ifngr1⁻/⁻Rag2⁻/⁻* mice of each condition, showing the whole right lobes. Scale bar: 3 mm. Data, except for (**c**) are representative of at least three independent experiments and are presented as the mean ± s.e.m. For statistical analysis, the following tests were used: (**a**–**c**) one-way ANOVA with Dunnett's multiple comparisons tests; (**e**) two-way ANOVA with Sidak's multiple comparisons tests. For (**a**, **b**, **c**–**e**) source data are provided as a Source Data file.

in these mice, as expected (Fig. 4e). The number of Siglec-F^low CD11c^+ macrophages tended to decrease but was not significant (Fig. 4e). These data demonstrated that ILC2s or ST2⁻KLRG1⁻ cells, or both, play a causal role in the development of PF in *Ifngr1⁻/⁻Rag2⁻/⁻* mice.

Because *Ifngr1⁻/⁻Il2rg⁻/⁻Rag2⁻/⁻* mice congenitally lack ILC2s and ST2⁻KLRG1⁻ cells, we next investigated whether the acquired elimination of ILC2s by neutralizing antibodies could prevent fibrosis. Because the expression of *Thy1* was limited to ILCs, including both ILC2s and ST2⁻KLRG1⁻ cells (Fig. 4f), we selected an anti-Thy-1 antibody for cell depletion. After checking that anti-Thy-1 antibodies completely depleted the CD45⁺lineage⁻Thy-1⁺ cells, *Ifngr1⁻/⁻Rag2⁻/⁻* mice were treated with anti-Thy-1 antibodies every 3 days for 6 weeks during the inflammatory phase (Fig. 4g), which led to complete suppression of PF (Fig. 4h and i), thereby indicating that ILCs are indispensable for the onset of PF.

## ILC2s and ILC3s subpopulations increase with PF progression

To determine which cells among ILCs primarily contribute to fibrosis, a more detailed subset analysis of ILCs was performed using scRNA-seq data shown in Fig. 3. Unsupervised clustering divided the ILCs into eight subclusters (Fig. 5a) with distinct differences in gene expression patterns (Supplementary Fig. 7a). Since there are numerous ILC subtypes, we determined whether each cluster belonged to ILC1s, ILC2s, or ILC3s using a previously published method that scores clusters

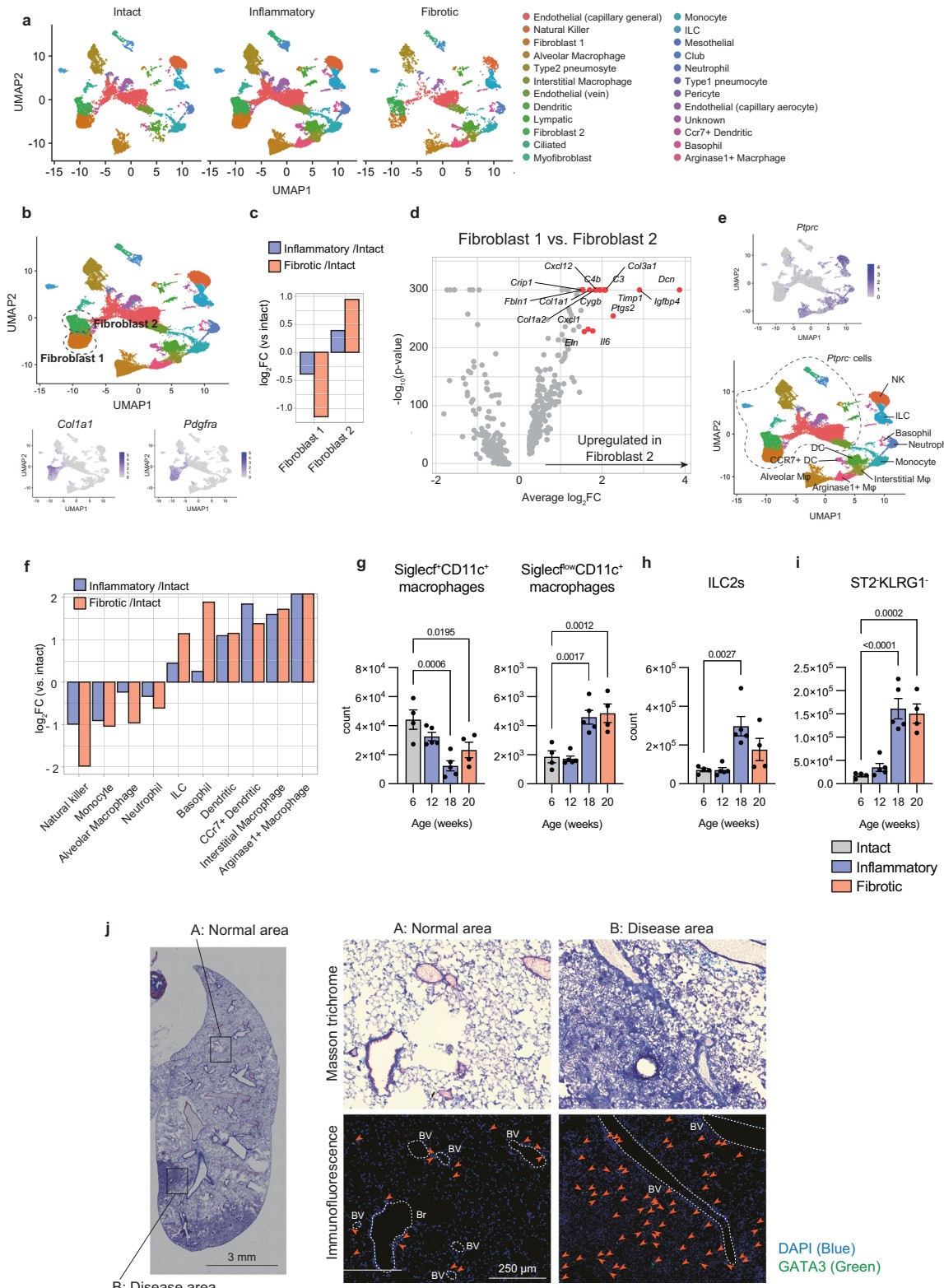

based on ILC-related gene expression (Fig. 5b)[22]. From these data, we identified clusters 0 and 7 as ILC1s; clusters 2, 3, 4, and 6 as ILC2s; and clusters 1 and 5 as ILC3s. The expression of *Tbx21*, *Gata3*, and *Rorc*, the master transcription factors of ILC1s, ILC2s, and ILC3s, respectively, supported this classification (Fig. 5c). Cluster 5 ILC3s showed elevated expression of *Ccr6*, *Cd4*, *Ccr7*, and *Tnfsf4*, which are signature genes for a LTi-like ILC3 phenotype, while cluster 1 ILC3s displayed *Ccr6* expression but lacked *Cd4* and *Ncr1* expression, indicative of an ex-

NKp46+ ILC3 phenotype[23] (Supplementary Fig. 7b and c). Composition analysis of the subpopulations in terms of progression phases showed that both ILC3s (cluster 1) and ILC2s (cluster 6) increased from the inflammatory phase (Fig. 5d). Based on the expression of *Thy1*, *Il1rl1* (IL-33 receptor), and *Klrg1* (Fig. 5e), cluster 1 ILC3s are likely to be CD45+lineage-Thy-1+ST2-KLRG1- cells, which increases in number from the inflammatory phase onward (Fig. 3i). Since cluster 1 ILC3s displayed a lower expression of ILC3-related genes than the typical cluster

**Fig. 3 | ILC2s and ST2⁻KLRG1⁻ cells increase with PF progression. a–f** scRNA-seq of whole lung cells from *Rag2⁻/⁻* mice (18 weeks old) and *Ifngr1⁻/⁻Rag2⁻/⁻* mice (7, 12, 16, 19, and 24 weeks old) (*n* = 2/group; females). **a** Unsupervised clustering of the combined data set plotted on UMAP (Uniform Manifold Approximation and Projection), divided by each phase, and colored according to the identified cell types. **b** All the cells of the combined data set are plotted on UMAP. The clusters defined as Fibroblast 1 and Fibroblast 2 are indicated by dotted circles. The lower panels show the expression of *Col1a1* and *Pdgfra*. **c** Relative abundance of Fibroblast 1 and Fibroblast 2 in *Ptprc⁻* clusters, compared to that in the intact phase. **d** Volcano plot of differentially expressed genes (|log₂[fold change]| > 0.25) between Fibroblast 1 and Fibroblast 2. Upregulated genes (log₂[fold change] > 1.5; *P* < 10⁻¹⁵⁰) are highlighted in red. **e** The upper panel shows the expression of *Ptprc*. The lower panel shows the identified immune cells. **f** Relative abundance of the indicated cell types in *Ptprc⁺* clusters, compared to the intact phase. **g–i** Flow cytometry analysis of the lungs of *Ifngr1⁻/⁻Rag2⁻/⁻* mice of different ages as indicated (6 and 20 weeks: *n* = 4/group; 12 and 18 weeks: *n* = 5/group; females). Quantification of the absolute number of indicated cells of lungs in each phase. **j** MT and immunofluorescence staining images of the whole right lung lobes of *Ifngr1⁻/⁻Rag2⁻/⁻* mice (24 weeks; male). The areas enclosed by the squares in the left panel are enlarged and shown in the right panels (A: normal area, and B: disease area). The arrows indicate Gata3-positive cells. The dotted lines indicate blood vessels or bronchi. BV, blood vessel; Br, bronchus. Blue, DAPI; Green, Gata3. Scale bars: 3 mm (left panel) or 250 μm (right panels). Data are representative of at least three independent experiments and are presented as the mean ± s.e.m. For statistical analysis, the following tests were used: **d** two-sided Wilcoxon Rank Sum test with the Bonferroni method. **g–i** one-way ANOVA with Dunnett's multiple comparisons tests. For (**g**, **h**, **i**) source data are provided as a Source Data file.

---

5 ILC3s (Fig. 5b), ST2⁻KLRG1⁻ cells (cluster 1 ILC3s) will be referred to as ILC3-like cells from hereon.

Given that both flow cytometry and scRNA-seq results indicated the involvement of ILC3-like cells in the pathogenesis of PF, we next depleted ILC3s by knocking out *Rorc* expression from *Ifngr1⁻/⁻Rag2⁻/⁻* mice, to validate the role of ILC3-like cells in PF. Flow cytometric analysis showed a significant decrease in the number and frequency of ILC3-like cells in *Ifngr1⁻/⁻Rorc^gfp/gfp^Rag2⁻/⁻* mice compared with that in *Ifngr1⁻/⁻Rag2⁻/⁻* mice (Fig. 5f). PF did not occur in *Ifngr1⁻/⁻Rorc^gfp/gfp^Rag2⁻/⁻* mice (Fig. 5g, h), indicating that ILC3-like cells are involved in the progression of PF in *Ifngr1⁻/⁻Rag2⁻/⁻* mice. Various analyses were performed to determine why the lack of *Rorc* inhibits fibrosis. And consequently, we found that *Ifngr1⁻/⁻Rorc^gfp/gfp^Rag2⁻/⁻* mice showed a drastic decrease in ILC2s in the BALF in the inflammatory phase, even though the proportion of ILC2s in the lung interstitium was unchanged (Fig. 5i). Since lung ILC2s are known to infiltrate alveoli upon activation in a mouse model of asthma and helminth infection[4], the absence of ILC2s in the BALF of *Ifngr1⁻/⁻Rorc^gfp/gfp^Rag2⁻/⁻* mice suggested that the activation of ILC2s is inhibited by *Rorc* deficiency. Since it is reported that *Rorc* deficiency did not affect the transcriptional landscape of ILC2s[24], it is more likely that ILC2 activation was suppressed exogenously by the lack of ILC3-like cells than by intrinsic *Rorc* deficiency. Taken together, these results suggested that the loss of ILC3s and the dysfunction of ILC2s may be responsible for the lack of fibrosis in *Ifngr1⁻/⁻Rorc^gfp/gfp^Rag2⁻/⁻* mice.

### IL-33-mediated activation of ILC2s is indispensable for PF progression

To investigate the involvement of ILC2s in PF, we focused on cluster 6 ILC2s, which appear in the lungs during the inflammatory phase of fibrosis. Compared to the other ILC2 clusters (clusters 2, 3, and 4), cluster 6 ILC2s expressed high levels of *Il13*, a fibrosis factor; *Il2ra*, an ILC2 activation marker; and *Il1rl1*, the most important factor for ILC2 activation in the lungs (Fig. 6a). Real-time quantitative PCR analysis demonstrated that the expression of *Il33* in the lungs increased in the fibrotic areas, but not normal areas, of the 22-week-old mice (Fig. 6b). Since we found that *Il1rl1* expression is restricted in ILC2s (Figs. 3e, 5e, and 6c), *Ifngr1⁻/⁻Rag2⁻/⁻Il33^gfp/gfp^* mice were generated to examine the importance of the IL-33-ILC2 axis in PF. *Il33* depletion in *Ifngr1⁻/⁻Rag2⁻/⁻Il33^gfp/gfp^* mice markedly suppressed both the spontaneous onset of fibrosis (Fig. 6d, e) and the infiltration of ILC2s in the BALF (Fig. 6f), thereby indicating that IL-33 is indispensable for the development of PF development in *Ifngr1⁻/⁻Rag2⁻/⁻* mice. Given that IL-33 is a powerful activator of ILC2s, and considering the high expression of its receptor in cluster 6 ILC2s, it is plausible to infer that IL-33 can induce the activation of these ILC2s in the lung, playing a crucial role in the development of PF. This finding is consistent with our observation that fibrosis does not occur in ILC-deficient mice (Fig. 4a).

To elucidate the cellular source of IL-33 in PF, we generated *Ifngr1⁻/⁻Rag2⁻/⁻Il33^gfp/+^* reporter mice. Almost 80% of IL-33/GFP⁺ cells were epithelial cells, a well-known source of IL-33 in the lungs during helminth infections and asthma[25]; however, IL-33/GFP expression was also observed in fibroblasts (Fig. 6g and Supplementary Fig. 8a). While IL-33/GFP expression was stable in the epithelial cells of 7-, 14-, and 22-week-old mice, it was elevated in the fibroblasts of 22-week-old mice (Supplementary Fig. 8b). Quantification of IL-33 expression in these cells sorted from *Ifngr1⁻/⁻Rag2⁻/⁻* mice showed an increase in IL-33 in fibroblasts associated with fibrosis (Fig. 6h and Supplementary Fig. 8c). Immunofluorescence staining for IL-33, GATA3 (a marker for ILC2s), PDGFRα (a marker for fibroblasts), and epithelial cell adhesion molecule (EpCAM) (a marker for epithelial cells) revealed that IL-33-positive cells are less common in the normal areas and are present with ILC2s in fibrotic areas (Fig. 6i and Supplementary Fig. 8d). The number of IL33⁺ cells and ILC2s in each ROI (region of interest) was quantified within sections (Fig. 6j and Supplementary Fig. 8e), which showed an increase in both IL33⁺ cells and ILC2s during fibrosis. These findings suggest that ILC2s activated by IL-33 from fibroblasts and epithelial cells are important for fibrosis development in *Ifngr1⁻/⁻Rag2⁻/⁻* mice.

### IL-33-activated ILC2s upregulate collagen production from fibroblasts

Since ILC2s are important for the induction of fibrosis, we wanted to clarify whether the relationship between ILC2s and fibroblasts is direct or indirect mediated by other cells. To address this question, we sorted ILC2s and fibroblasts from WT mice (Supplementary Fig. 9) and co-cultured these cells under stimulation with three different cytokines (Fig. 7a). IL-2 and IL-7 were used to maintain the survival of ILC2s, while IL-33 was used to activate ILC2s. Sirius Red staining, which stains collagen red, was performed after co-culture. Collagen production did not change in fibroblasts co-cultured with non-activated ILC2s, but appeared to be enhanced in fibroblasts co-cultured with activated ILC2s (Fig. 7a). To quantitatively analyze collagen production, the Sirius Red dye was eluted from fibroblasts and the absorption rate was measured. The absorption rate increased in the presence of IL-33-activated ILC2s (Fig. 7b), indicating that ILC2s directly induced collagen production in fibroblasts.

### ILC2s from individuals with IPF show a similar phenotype to ILC2s from *Ifngr1⁻/⁻Rag2⁻/⁻* mice

Finally, to test whether ILC2-mediated fibrosis development due to the lack of IFNγ signaling occurs in humans, we analyzed ILC2s in individuals with IPF. We performed RNA-sequencing (RNA-seq) analysis of peripheral blood ILC2s sorted from 12 healthy volunteers and 19 individuals with IPF (Supplementary Fig. 10), to understand whether the disease mechanisms found in mice also exist in humans. ILC2s of individuals with IPF were distinctly separated from those of healthy controls using principal component analysis (PCA) (Fig. 8a). Gene Ontology analysis of the highly expressed genes in ILC2s of individuals with IPF, in comparison to those in the controls, resulted in the enrichment of extracellular matrix-related terms in IPF-derived ILC2s (Fig. 8b). We then comprehensively examined the expression of genes

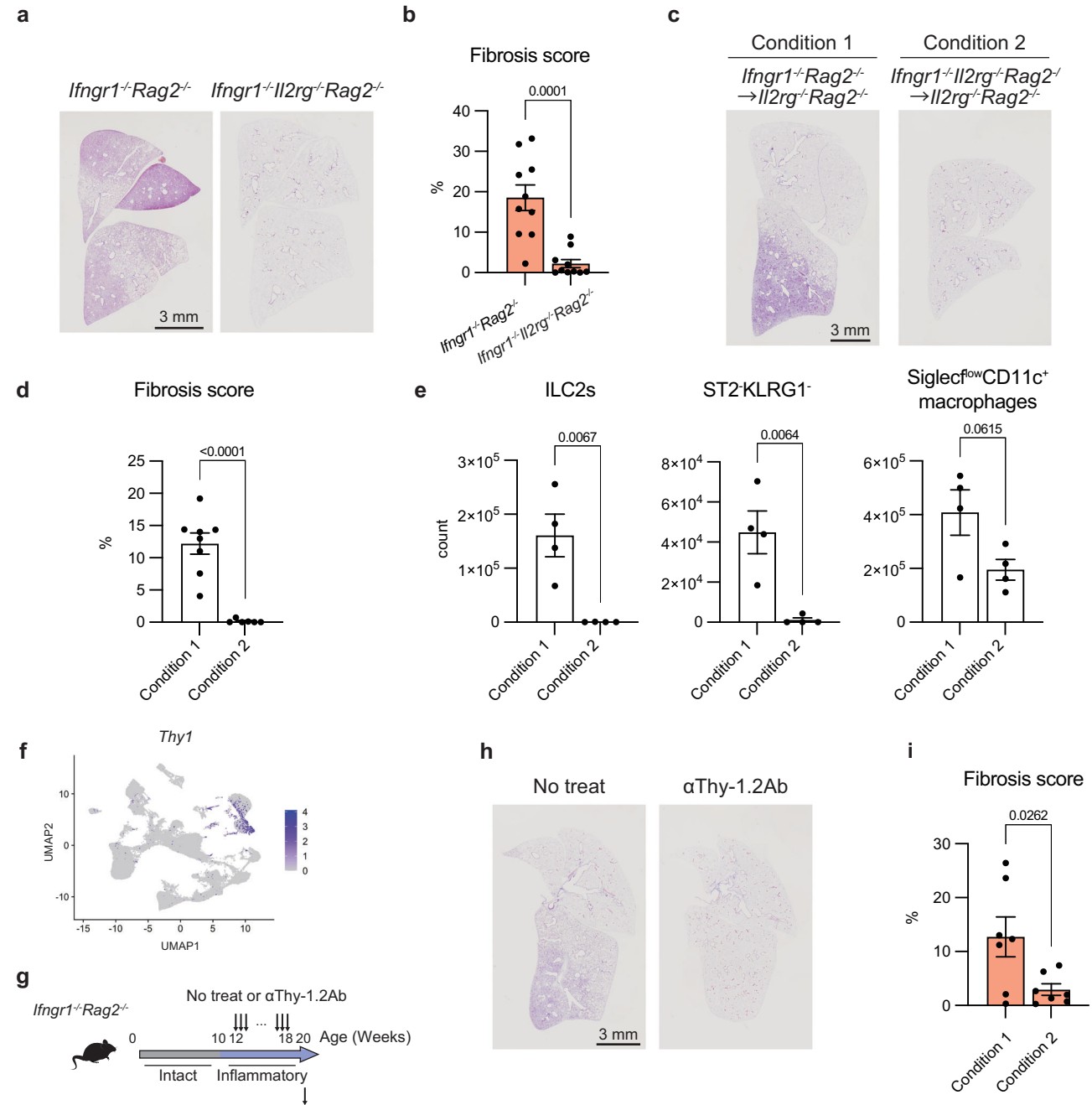

**Fig. 4 | ILCs are indispensable for PF progression. a** Representative MT-stained images of lung tissue sections of the indicated mice, showing the whole right lobes (*Ifngr1⁻/⁻Rag2⁻/⁻* mice: 27 weeks; male, *Ifngr1⁻/⁻Il2rg⁻/⁻Rag2⁻/⁻* mice: 27 weeks; female). Scale bar: 3 mm. **b** Fibrosis scores calculated using MT-stained images of lung tissue sections from the indicated mice (*Ifngr1⁻/⁻Il2rg⁻/⁻Rag2⁻/⁻* mice: n = 10/group [4 at 21 weeks, 1 at 23 weeks, 3 at 27 weeks, 1 at 28 weeks, 1 at 46 weeks]; 9 females and 1 male, *Ifngr1⁻/⁻Rag2⁻/⁻* mice: n = 10/group [1 at 22 weeks, 5 at 23 weeks, 1 at 25 weeks, 2 at 26 weeks, 1 at 27 weeks]; 5 females and 5 males). **c**−**e** Bone marrow transfer (BMT) experiment. See the Methods section and Supplementary Fig. 6 for details (n = 4/group; females). **c** Representative MT-stained images of lung tissue sections of mice with each condition, showing the whole right lobes. Scale bar: 3 mm. **d** Fibrosis scores were calculated using MT-stained images of lung tissue sections of mice from each condition. The results of four experiments were integrated to calculate the

scores. **e** Quantification of the absolute number of the indicated cells in the lungs of mice with each condition by flow cytometry. **f** The expression of *Thy-1* in all lung cells. **g**−**i** Depletion of ILCs using anti-Thy-1 antibodies. **g** Schematic of the experiment. *Ifngr1⁻/⁻Rag2⁻/⁻* mice were intraperitoneally administered anti-Thy-1 antibodies (clone: 30H12) (200 µg/head per shot) every 3 days for 6 weeks (12 weeks old at the start of administration, n = 3/group; females). **h** Representative MT-stained images of lung tissue sections of mice in each condition, showing the whole right lobes. Scale bar: 3 mm. **i** Fibrosis scores were calculated using MT-stained images of lung tissue sections of mice in each condition. The results of two experiments were integrated to calculate the scores. Data are representative of at least three independent experiments and are presented as the mean ± s.e.m. For statistical analysis, the following tests were used: (**b**−**d**, **e**−**i**) two-tailed unpaired Student's *t*-tests. For (**b**−**d**, **e**−**i**) source data are provided as a Source Data file.

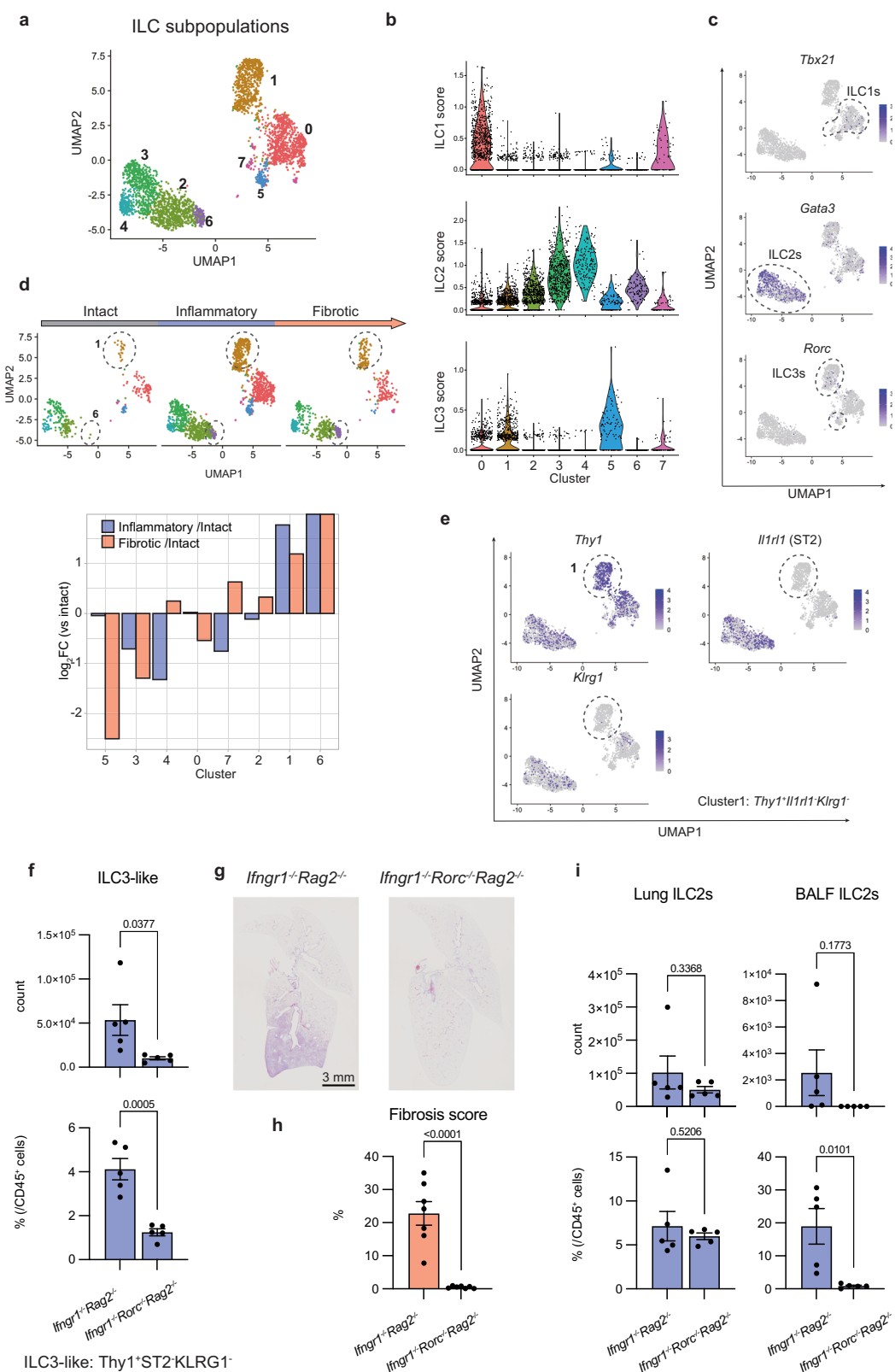

that promote PF[26] and found that many of them are upregulated in IPF-derived ILC2s (Fig. 8c). The expression of *IL5*, *IL13*, and *IL1RL1* was significantly upregulated in ILC2s from individuals with IPF, as compared to that in ILC2s from healthy volunteers, indicating ILC2 activation in individuals with IPF (Fig. 8d). Most importantly, there was significantly decreased expression of *IFNGR1* in ILC2s from individuals with IPF (Fig. 8d), suggesting that IFNγ-mediated suppression of ILC2s

is also attenuated in individuals with IPF. These data indicated that ILC2s are also activated in individuals with IPF, consistent with the mechanism observed in *Ifngr1⁻/⁻Rag2⁻/⁻* mice.

## Discussion
In this study, we found that PF develops spontaneously in *Ifngr1⁻/⁻Rag2⁻/⁻* mice, thereby demonstrating its applicability as a model of PF that is

**Fig. 5 | ILC2s and ILC3s subpopulations increase with PF progression.**
**a**–**e** Clusters identified as ILCs were extracted from all lung cells in the scRNA-seq data shown in Fig. 3, on which unsupervised clustering was performed. **a** Unsupervised clustering of ILCs plotted on UMAP and colored according to each cluster. **b** Violin plots of the ILC1, ILC2, and ILC3 scores of each cluster, calculated from the average expression levels of signature genes of each population. **c** The expression of *Tbx21*, *Gata3*, and *Rorc* in ILCs. **d** The upper panel shows an unsupervised clustering of ILCs plotted on UMAP divided by each phase. The dotted circles show clusters 1 and 6. The lower panel shows the relative abundance of each cluster in ILCs compared to the intact phase. **e** The expression of *Thy-1*, *Il1rl1*, and *Klrg1* in ILCs. **f** Quantification of the absolute number and ratio of lung ILC3-like cells in the indicated mice by flow cytometry (17 weeks for *Ifngr1^-/-^Rag2^-/-^* mice and 19 weeks old for *Ifngr1^-/-^Rorc^-/-^Rag2^-/-^* mice, *n* = 5/group; males). **g** Representative MT-stained

images of lung tissue sections of the indicated mice, showing the whole right lobes. Scale bar: 3 mm. **h** Fibrosis scores were calculated using MT-stained images of lung tissue sections from the indicated mice (*Ifngr1^-/-^Rorc^-/-^Rag2^-/-^* mice: *n* = 7/group [2 at 21 weeks, 2 at 24 weeks, 2 at 34 weeks, 1 at 37 weeks]; 6 females and 1 male, *Ifngr1^-/-^Rag2^-/-^* mice: *n* = 7/group [3 at 21 weeks, 1 at 22 weeks, 2 at 21 weeks, 1 at 22 weeks, 4 at 23 weeks]; 3 females and 4 males). **i** Quantification of the absolute number and ratio of both lung and BALF ILC2s of the indicated mice by flow cytometry (17 weeks for *Ifngr1^-/-^Rag2^-/-^* mice and 19 weeks for *Ifngr1^-/-^Rorc^-/-^Rag2^-/-^* mice; *n* = 5/group; males). Data are representative of at least three independent experiments and are presented as the mean ± s.e.m. For statistical analysis, the following tests were used: (**f**–**h**, **i**) two-tailed unpaired Student's *t*-tests. For (**f**–**h**, **i**) source data are provided as a Source Data file.

not exogenously induced. It has been reported that PF-like symptoms spontaneously occur in numerous mouse strains >2 years (at a rate of 2% in C57BL/6)[27]. Additionally, the incidence rate rises to 20% in *Rag2^-/-^* mice[5]. Notably, in our mouse model, the deletion of *Ifngr1* in *Rag2^-/-^* background mice dramatically increased the incidence of PF. This finding sheds light on the association between IFNγ and fibrosis, a topic that has been extensively discussed in the literature. It has been reported that IFNγ suppresses fibroblasts and decrease collagen production[28], and that the amount of IFNγ is reduced in the serum of individuals with IPF[29]. Building upon these findings, our study provides a mechanistic insight by proposing ILC2s as a target of IFNγ during PF, where a lack of IFNγ exacerbates fibrosis by activating ILC2s. In our previous studies on the mechanism of ILC2 suppression, we compared various cytokines that suppress ILC2s and ultimately found that IFNγ did so to the greatest degree[4,30]. Although IFNγ is a well-known cytokine produced during viral and intracellular bacterial infections, the results of this study suggest that IFNγ is produced constantly, even in the absence of foreign antigens, and may prevent excessive ILC2 activation. The inhibition of PF development in ILC3-deficient mice was unexpected; it is possible that IFNγ constitutively suppresses both ILC2s and ILC3s, since there are reports that IFNγ suppresses ILC3s[31]. It is likely that ILC2s are downstream of ILC3s in the development of PF in *Ifngr1^-/-^Rag2^-/-^* mice, since depletion of ILC3s prevents the infiltration of activated ILC2s in the alveoli. While this study focused on the activation of ILC2s during the inflammatory phase and the interaction between ILC2s and fibroblasts during the chronic phase, we are also carrying out a future study to determine how the activation of ILC2s or ILC3s is triggered in the early stages of fibrosis.

The importance of PF studies using *Ifngr1^-/-^Rag2^-/-^* mice is further supported by the progression pattern of fibrosis. In human IPF, fibrosis typically initiates from the subpleural lower lobe of the lung interstitium[28], suggesting the presence of factors triggering fibrosis on the pleural side. Although bleomycin-induced fibrosis models have yielded valuable insights, they do not fully replicate the actual pathophysiology of PF since fibrosis begins along the route of administration, such as the bronchial side[11]. Conversely, in most cases, fibrosis in *Ifngr1^-/-^Rag2^-/-^* mice initiates from the pleural side, as evidenced by the MT staining images presented herein. Analysis of these mice may provide valuable insights into the mechanism by which IPF develops from the pleural side. In addition, despite previous reports indicating that lung ILC2s predominantly reside in the adventitial cuffs of lung vessels and airways during helminth infection[21], our findings demonstrate that the distribution of ILC2s and the expression of IL-33 completely overlap with fibrotic regions in *Ifngr1^-/-^Rag2^-/-^* mice. This observation suggests a potential involvement of ILC2s and IL-33 in the development of fibrosis in this mouse model.

The activation of ILC2s is induced by various factors, including IL-2, IL-25, IL-9, and IL-33. Among these factors, IL-33 is the strongest ILC2 activator in the lung[1,4]. It has been reported that defects in the IL-33 receptor ameliorate the disease state in a bleomycin-induced PF mouse model[32]. Considering that bleomycin-induced damage to

epithelial cells leads to IL-33 secretion[33], the exogenously induced IL-33 should play a crucial role in the progression of fibrosis in bleomycin-induced PF. Remarkably, fibrosis in *Ifngr1^-/-^Rag2^-/-^* mice was completely abolished by the loss of IL-33, indicating that endogenous IL-33 production occurs independently of exogenous epithelial injury and contributes to fibrosis development. Flow cytometric analysis detected IL-33 expression in both fibroblasts and epithelial cells, with no significant difference in expression levels observed between the intact, inflammatory, and chronic phases in epithelial cells. In contrast, IL-33 expression in fibroblasts increased from the inflammatory phase and became more pronounced during the chronic phase. The localization of ILC2s was observed in areas exhibiting active fibroblast proliferation, suggesting that the interaction between fibroblasts and ILC2s may be involved in the chronic and irreversible progression of the disease. The present study did not clarify whether fibroblast-derived or epithelial cell-derived IL-33 is more important for fibrosis, but it will be important to clarify this point in future studies using conditional knockout mice. Further studies are also needed to determine if there are differences in the sources of IL-33 that trigger fibrosis and those that contribute to the chronicity of fibrosis.

We observed an increased number of Siglec-F^low^CD11c^+^ macrophages and ILCs as the disease progressed. Macrophages are known to promote fibrosis by producing TGF-β[18,34]. Although a detailed analysis of macrophages was not performed in this study, since ILC2s affect macrophage maturation by producing IL-4 and IL-13[35], there may be a mechanism by which ILC2s can exacerbate fibrosis by affecting macrophages.

One of the major findings of this study was that ILC2s directly induced collagen expression in fibroblasts using a co-culture system. Immunofluorescence staining also revealed close contact between ILC2s and fibroblasts. While we have confirmed a partial reduction in collagen production by fibroblasts through IL-13 neutralization, it is important to note that fibrosis may result from the interplay of multiple factors rather than a single isolated factor. Identifying and characterizing these factors will be the focus of future studies.

Given the similar gene expression pattern in ILC2s between our mouse model and individuals with IPF, we speculate that the pathological mechanisms in *Ifngr1^-/-^Rag2^-/-^* mice may also exist in humans. This notion is supported by reports of increased ILC2 counts in the BALF of individuals with IPF[14] and higher ILC2 counts in peripheral blood in individuals with IPF with poor prognoses[36], aligning with our findings. IIPs are classified into nine subgroups, and IPF is the most common and severe disease among them. While we have validated our findings in individuals with IPF, demonstrating similarities with *Ifngr1^-/-^Rag2^-/-^* mice, we have not validated our results in patients with other IIPs with PF. The extent to which *Ifngr1^-/-^Rag2^-/-^* mice can contribute to our understanding of other IIPs remains to be determined. However, it is worth noting that a spontaneous mouse model of pulmonary fibrosis offers valuable insights that can significantly enhance our understanding of fibrosis development. Furthermore, the findings of our study demonstrate the essential role of ILCs in the onset of PF.

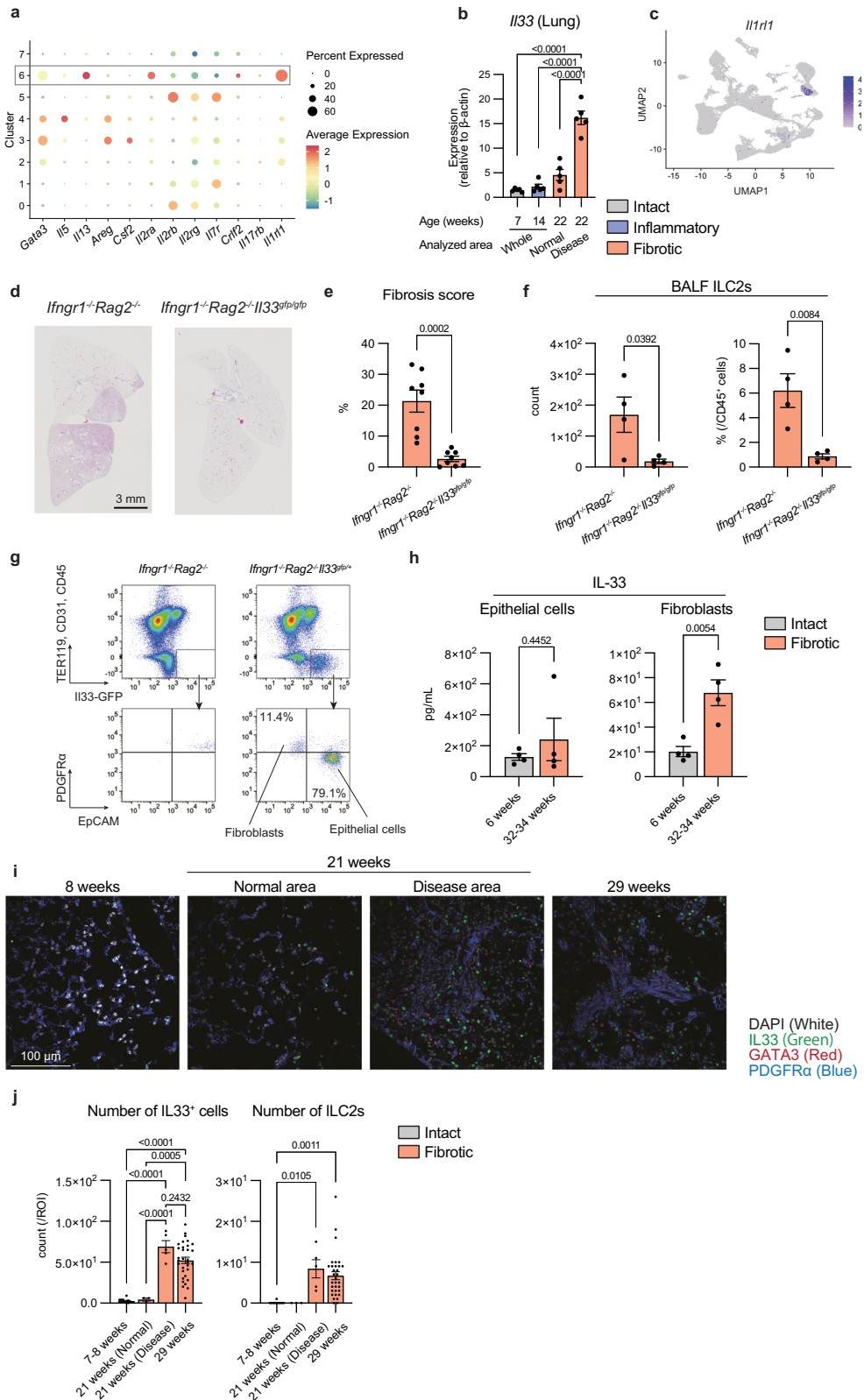

Therapeutic interventions aimed at targeting ILCs during the initial stages of PF may mitigate the progression of fibrosis.

## Methods
### Mice
All mice used in this study had a C57BL/6 background and were maintained under specific pathogen-free conditions in an animal facility at the RIKEN Center for Integrative Medical Sciences (Kanagawa, Japan). Mice were housed in an environment with a 12-h light/dark cycle, maintaining an ambient temperature between 21 °C and 25 °C, and humidity levels controlled at 40–60%. All experiments were approved by the Animal Care and Use Committee of RIKEN and performed in accordance with the institutional guidelines. All mice used in this study were euthanized under anesthesia.

**Fig. 6 | IL-33-mediated activation of ILC2s is indispensable for PF progression.** **a** Dot plot of sub-clustered ILC subsets showing the ILC2-related genes. Cluster 6 is highlighted. **b** mRNA expression of *Il33* in the lungs of *Ifngr1⁻/⁻Rag2⁻/⁻* mice quantified by RT-qPCR. See the Methods section for details (*n* = 5/group; males). **c** The expression of *Il1rl1* in all lung cells. **d** Representative MT-stained images of lung tissue sections, showing the whole right lobes. Scale bar: 3 mm. **e** Fibrosis scores were calculated using MT-stained images of lung tissue sections. The results of several experiments were combined. See the Methods section for details (*Ifngr1⁻/⁻Rag2⁻/⁻Il33ᵍᶠᵖ/ᵍᶠᵖ* mice: *n* = 8/group [2 at 22 weeks, 1 at 23 weeks, 1 at 24 weeks, 3 at 25 weeks, 1 at 36 weeks]; 4 females and 4 males, *Ifngr1⁻/⁻Rag2⁻/⁻* mice: *n* = 8/group [2 at 21 weeks, 1 at 22 weeks, 4 at 23 weeks, 1 at 27 weeks]; 4 females and 4 males). **f** Quantification of the absolute number and ratio of BALF ILC2s by flow cytometry (*n* = 4/group [*Ifngr1⁻/⁻Rag2⁻/⁻* mice: 4 at 24 weeks, *Ifngr1⁻/⁻ Rag2⁻/⁻Il33ᵍᶠᵖ/ᵍᶠᵖ* mice: 1 at 24 weeks, 3 at 23 weeks]; females). **g** Representative flow cytometry plot of the lung cells. **h** Quantification of IL-33 in both epithelial cells and fibroblasts sorted from the lungs of *Ifngr1⁻/⁻Rag2⁻/⁻* mice of different ages using ELISA (*n* = 4/group; males). **i, j** Immunofluorescence staining of the lungs of *Ifngr1⁻/⁻Rag2⁻/⁻* mice (*n* = 1/group; males). **i** Representative immunofluorescence staining images of the lungs of *Ifngr1⁻/⁻Rag2⁻/⁻* mice. White, DAPI; Green, IL-33; Red, Gata3; Blue, PDGFRα. Scale bar: 100 μm. **j** Quantification of the number of IL33⁺ cells and ILC2s (defined by GATA3⁺ expression) per image. See Supplementary Fig. 8e for details on the analysis process. Data are representative of at least three independent experiments and are presented as the mean ± s.e.m. For statistical analysis, the following tests were used: (**e, f–h**) two-tailed unpaired Student's *t*-tests; (**b–j**) one-way ANOVA with Tukey's multiple comparisons tests. For (**b–e, f–h–j**) source data are provided as a Source Data file.

**a**

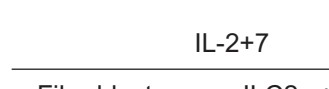

IL-2+7

Fibroblasts    ILC2s + Fibroblasts

IL-2+7+33

Fibroblasts    ILC2s + Fibroblasts

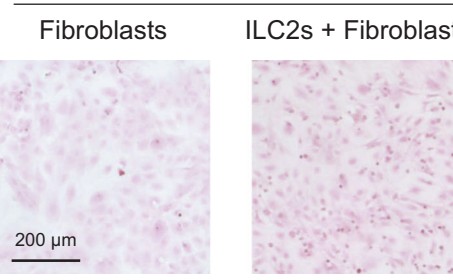

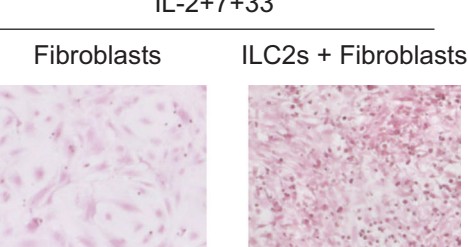

200 μm

**b** Absorption rate of eluted Sirius red

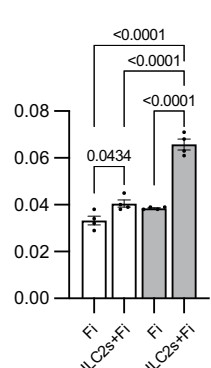

**Fig. 7 | IL-33-activated ILC2s upregulate collagen production from fibroblasts.** **a, b** ILC2s and fibroblasts were sorted from WT mice on day 0 and were co-cultured (ILC2s, 5000 cells/well; fibroblasts, 20,000 cells/well; IL-2, -7, and -33, 10 ng·mL⁻¹) under the indicated conditions. On day 5, ILC2s were washed out, and fibroblasts were stained with Sirius red. Cells sorted from 10 mice were combined and seeded into four separate wells. This experiment was repeated multiple times, consistently yielding similar results in each iteration. **a** Representative images of Sirius red staining in the indicated conditions. Scale bar: 200 μm. **b** Absorption rate of the eluted Sirius red (*n* = 4 wells /condition). Fi, fibroblasts. Data are representative of at least three independent experiments and are presented as the mean ± s.e.m. For statistical analysis, the following tests were used: (**b**) one-way ANOVA with Tukey's multiple comparisons tests. For (**b**) source data are provided as a Source Data file.

WT C57BL/6 N mice were purchased from Charles River Laboratories Japan (Kanagawa, Japan) or CLEA (Tokyo, Japan). B6- *Rag2⁻/⁻* (stock no. RAGN12) and *Il2rg⁻/⁻Rag2⁻/⁻* mice (stock no. 4111) were purchased from Taconic Bioscience Japan (Tokyo, Japan). *Ifngr1⁻/⁻* mice (stock no. 003288) were purchased from Jackson Laboratory and crossed with *Rag2⁻/⁻* mice to generate *Ifngr1⁻/⁻Rag2⁻/⁻* mice. *Ifng⁻/⁻* mice were provided by Yoichiro Iwakura (Tokyo University of Science, Chiba, Japan)[37], *Il33ᵍᶠᵖ/ᵍᶠᵖ* mice were provided by Susumu Nakae (Hiroshima University, Hiroshima, Japan)[38], and *Rorcᵍᶠᵖ/ᵍᶠᵖ* mice were provided by Sidonia Fagarasan (RIKEN, Yokohama, Japan)[39]. *Ifng⁻/⁻* mice were crossed with *Rag2⁻/⁻* mice to generate *Ifng⁻/⁻Rag2⁻/⁻* mice. *Rorcᵍᶠᵖ/ᵍᶠᵖ* mice were crossed with *Rag2⁻/⁻* mice to generate *Rorcᵍᶠᵖ/ᵍᶠᵖRag2⁻/⁻* mice. *Ifngr1⁻/⁻Rag2⁻/⁻* mice were crossed with *Il2rg⁻/⁻Rag2⁻/⁻*, *Rorcᵍᶠᵖ/ᵍᶠᵖRag2⁻/⁻*, or *Il33ᵍᶠᵖ/ᵍᶠᵖ* mice to generate *Ifngr1⁻/⁻Il2rg⁻/⁻Rag2⁻/⁻*, *Ifngr1⁻/⁻Rorcᵍᶠᵖ/ᵍᶠᵖRag2⁻/⁻*, *Ifngr1⁻/⁻Rag2⁻/⁻Il33ᵍᶠᵖ/⁺*, and *Ifngr1⁻/⁻Rag2⁻/⁻Il33ᵍᶠᵖ/ᵍᶠᵖ* mice.

We initially verified the development of fibrosis in both male and female mice (Fig. 1c, d). Consequently, both sexes were included in all subsequent experiments. Further information regarding the sex of the mice in each dataset is provided in the figure legends and the Source data.

**Micro-computed tomography (CT) lung imaging**
The mice were sacrificed at the age of 7, 13, 17, and 37 weeks old, and an 18 G plastic cannula was placed in their trachea. To obtain clear pictures of the lungs, the lungs were filled with 500 μL air using a 1 mL syringe via the cannula. The lungs were then scanned to gain axial images using micro-CT (ScanXmate-RX, Comscantechno, Kanagawa, Japan).

**Measurement of collagen amount**
Lungs were isolated immediately after sacrificing the mice. Specifically, samples of 22-week-old *Ifngr1⁻/⁻Rag2⁻/⁻* mice were collected from both the normal and disease areas, whereas all other samples were obtained from the normal area due to the absence of any lesions. The disease area was defined as a visibly distinct white region, as depicted in Fig. 1a. The lungs were then homogenized using a BioMasher homogeniser (Nippi, Tokyo, Japan), and the resulted homogenate was washed with phosphate-buffered saline (PBS) to remove blood. Each homogenate was incubated overnight in 1 mL of 0.1 μg·mL⁻¹ pepsin (Sigma-Aldrich, St. Louis, MO, USA), and dissolved in 0.5 M acetic acid (Nacalai Tesque, Kyoto, Japan) to extract collagen. Collagen levels were assessed using the Sircol Collagen Assay Kit (#S1000, Biocolor, Carrickfergus, UK) following the manufacturer's instructions. The collagen volume was normalized to the corresponding lung wet weight.

**SP-D measurements using enzyme-linked immunosorbent assay (ELISA)**
Blood of *Ifngr1⁻/⁻Rag2⁻/⁻* mice was collected in 1.5-mL tubes containing a serum-separating medium (Bloodsepar, Immuno-Biological Laboratories, Gunma, Japan), and left for 30 min at 24 °C. Serum was recovered by means of centrifugation at 15,000 × *g* for 15 min at 4 °C. Serum SP-D levels were determined by performing ELISA using a Quantikine

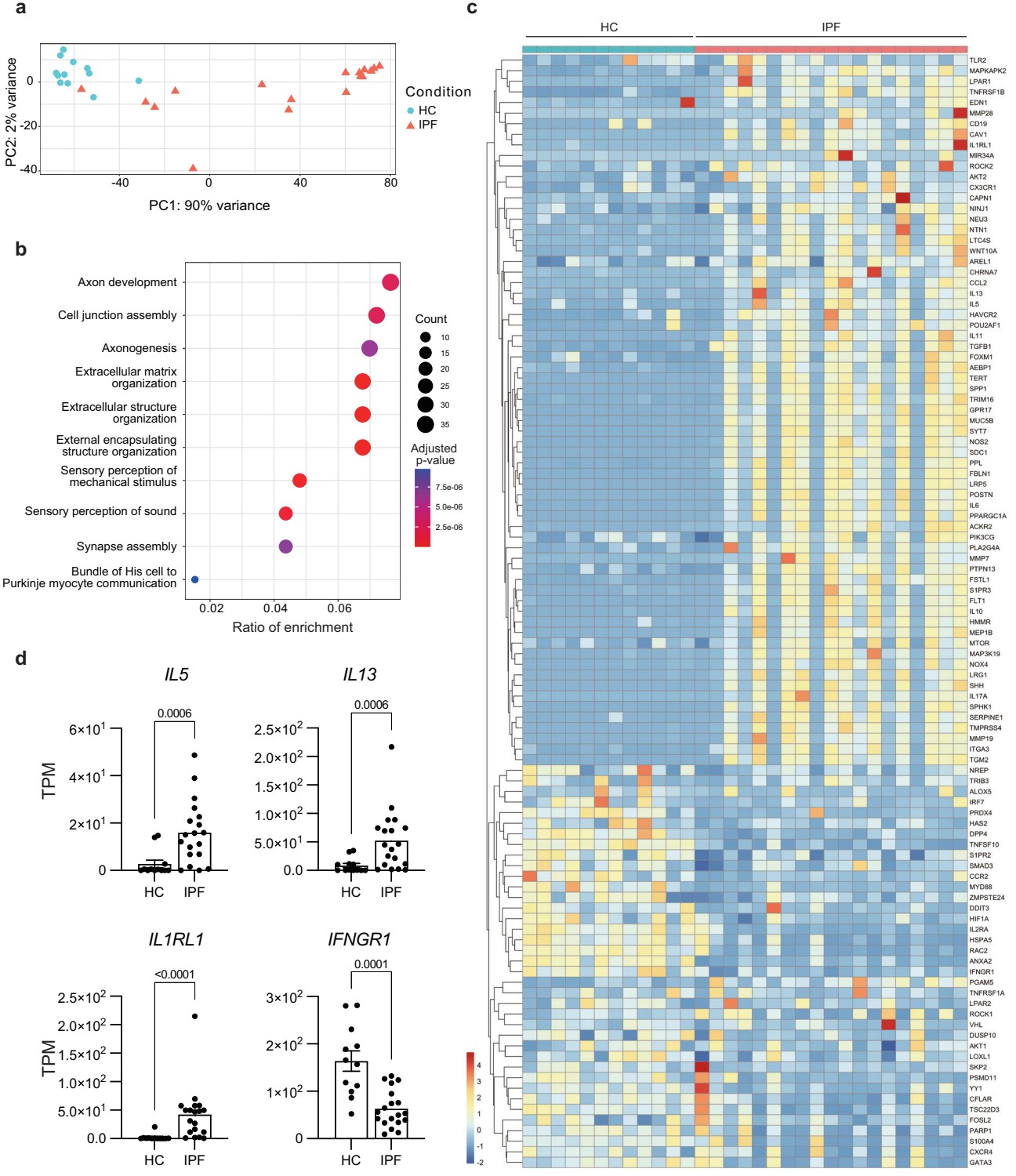

**Fig. 8 | ILC2s from individuals with IPF show a similar phenotype to ILC2s from *Ifngr1⁻/⁻Rag2⁻/⁻* mice.** **a**–**d** RNA-seq of peripheral blood ILC2s sorted from 12 healthy volunteers and 19 patients with IPF. **a** Principal component analysis (PCA) of each sample. HC: healthy control. **b** Gene ontology enrichment analysis of the highly expressed genes in ILC2s from patients with IPF compared to those from healthy controls (log₂[fold change] > 5; $P < 10^{-25}$). **c** Heatmap of z-scores of the transcripts per kilobase million (TPM values) of fibrosis-related genes. HC: healthy control.

**d** The TPM values of the indicated genes of ILC2s from healthy controls and patients with IPF. HC: healthy control. Data are presented as the mean ± s.e.m. For statistical analysis, the following tests were used: (**b**) over representation analysis (ORA), which corresponds to a one-sided Fisher's exact test, with the Benjamini−Hochberg method. **d** Two-sided Mann-Whitney test. For (**d**) source data are provided as a Source Data file.

Kit (#MSFPD0, R&D Systems, Minneapolis, MN, USA) following the manufacturer's instructions. The absorption rate was measured at 450 nm (reference wavelength, 570 nm) using a microplate reader (iMark, Bio-Rad, CA, USA).

## Measurement of oxygen saturation

The mice were anesthetized with an intraperitoneal injection of mixed anesthetic agents[40] (0.3 mg·kg$^{-1}$ medetomidine, 4.0 mg·kg$^{-1}$ midazolam, and 5.0 mg·kg$^{-1}$ butorphanol), and the fur around their necks was shaved. Next day, the mice were anesthetized with inhalation of isoflurane, after which a throat sensor was attached to the neck and oxygen saturation was measured following the manufacturer's instructions (MouseOx PLUS, Primetech, Tokyo, Japan).

## Respiratory function test

Mice were anesthetized with an intraperitoneal injection of mixed anesthetic agents and inhalation of isoflurane. Then, an 18 G plastic cannula was placed in the trachea. The mice were connected to a ventilator (FinePointe Resistance & Compliance, Primetech) via a cannula, after which static compliance ($C_{st}$) was measured following the manufacturer's instructions. $C_{st}$ is defined as

$$C_{st} = TV/(\text{Plateau pressure} - PEEP)$$

where $C_{st}$ is static compliance (mL·cm H$_2$O$^{-1}$), TV is the tidal volume (mL), and PEEP is the positive end-expiratory pressure (cm H$_2$O). PEEP was set to 2 cm H$_2$O.

## In vivo administration of corticosteroids

Osmotic minipumps (Alzet, Cupertino, CA, USA), which released 3 mg·kg$^{-1}$ dexamethasone (Sigma-Aldrich) per day, were implanted into the subcutaneous pockets under the back skin of the mice either from 14 to 18 weeks (inflammatory phase) or from 23 to 28 weeks old (fibrotic phase). Then, the mice were sacrificed and analyzed.

## Bone marrow transplantation

Whole bone marrow cells were collected from either *Ifngr1$^{-/-}$Rag2$^{-/-}$* or *Ifngr1$^{-/-}$Il2rg$^{-/-}$Rag2$^{-/-}$* mice. Both ends of the limb bones were cut, and bone marrow cells were flushed out using Hank's balanced salt solution containing 10 vol% heat-inactivated fetal calf serum (FCS) with an 18 G plastic cannula and 10 mL syringe. After lysing the red blood cells with Ammonium-Chloride-Potassium (ACK) lysing buffer, the bone marrow cells were intravenously transferred into 12-week-old *Il2rg$^{-/-}$Rag2$^{-/-}$* mice ($1 \times 10^7$ cells/head suspended in 100 μL phosphate-buffered saline [PBS]) without whole-body irradiation. The mice were sacrificed and analyzed at 21–31 weeks after transplantation.

## In vivo administration of anti-Thy-1.2 antibodies

We intraperitoneally administered anti-Thy-1.2 monoclonal antibodies (mAbs) (#BE0066; 30H12, Bioxcell, Lebanon, NH, USA) to the mice at a dose of 200 μg/head dissolved in 200 μL PBS every 3 days for 6 weeks, and then the mice were sacrificed and analyzed.

## Preparation of cell suspensions from lung tissues

BALF cells were collected by gently washing the lungs with Hank's balanced salt solution containing 10 vol% FCS using an 18 G plastic cannula and a 1 mL syringe. After collecting the BALF cells, the lungs were removed and minced with scissors in Hank's balanced salt solution containing 10 vol% FCS. The minced lungs were incubated with Liberase (#5401127001, Roche, Basel, Switzerland; final concentration: 50 μg·mL$^{-1}$) and DNase I (#10104159001, Roche; final concentration: 1 μg·mL$^{-1}$) for 45 min, at 37 °C. The digested lungs were further dissociated with gentleMACS (Miltenyi Biotec, Bergisch Gladbach, Germany), and the isolated cells were collected by passing through a 40-μm cell strainer. After lysing the red blood cells with ACK lysing buffer, immune cells were suspended in 30 vol% Percoll PLUS (#17-5445-01, GE Healthcare, Chicago, IL, USA) and centrifuged at $800 \times g$, 24 °C for 30 min, to remove epithelial cells. The resulting pellets were used for subsequent analysis after passing through a 37-μm filter. For epithelial cells and fibroblasts, the process of cell separation using Percoll was not performed.

## Antibodies for flow cytometry analysis

For flow cytometry analysis, the following antibodies were purchased from BD Biosciences (Franklin Lakes, NJ, USA): mAbs specific for mouse CD3ε (#553060; 145-2C11), CD4 (#553728; GK1.5), CD11c (#553800, #558079, #562782; HL3), CD19 (#553784; 1D3), Gr-1 (#553125; RB6-8C5), CD45.2 (#553772, #560696, #562895, #560694; 104), Thy-1.2 (#561616; 53-2.1), T1/ST2 (#566311; U29-93), KLRG1 (#583595; 2F1), and Siglec-F (#562680; E50-2440), which were conjugated to either biotin or a fluorochrome; fluorochrome-conjugated mAbs specific for human CD127 (#563165; hIL-7R-M21); and fluorochrome-conjugated streptavidin (#551419, #554063, #554067). The following antibodies were purchased from Thermo Fisher Scientific (Waltham, MA, USA): mAbs specific for mouse FcεRIα (#13-5898-85; MAR-1), F4/80 (#13-4801-85; BM8), and KLRG1 (#25-5893-82; 2F1), which were conjugated to either biotin or a fluorochrome. The following antibodies were purchased from BioLegend (San Diego, CA, USA): mAbs specific for mouse F4/80 (#123118; BM8), CD31 (#102404; 390), CD45 (#103104; 30-F11), EpCAM (#118216, #118204; 78.8), CD140α (#135923, #135906; APA5), which were conjugated with a fluorochrome; mAbs specific for human CD3 (#300424; UCHT1), CD4 (#300526; RPA-T4), CD14 (#325614; HCD14), CD16 (#302026; 3G8), CD19 (#302226; HIB19), FcεRIα (#334630; AER-37), CD45 (#304028; HI30), CD161 (#339916; HP-3G10), and CRTH2 (#350118; BM16), which were all conjugated with a fluorochrome. The following antibodies were purified from hybridoma culture supernatants in our laboratory: mAbs specific for mouse CD8α (53-6.7), NK1.1 (PK136), and TER119 (TER119), which were conjugated to biotin and purified CD16/CD32 (2.4G2). Zombie dyes (#423113, #423109, #423101, BioLegend) or propidium iodide (Sigma-Aldrich) were used to detect dead cells following their respective manufacturer's instructions.

## Flow cytometry analysis

Cells were collected as described above and stained with Zombie dyes (#423113, #423109, BioLegend) to distinguish live cells from dead cells, following the manufacturer's instructions. Then, the cells were centrifuged at 300 g, 4 °C for 3 min. After removing the supernatant, 50 μL of primary mAbs suspended in Hank's balanced salt solution containing 10 vol% FCS was added. The cells were incubated at 4 °C for 20 min, and washed with Hank's balanced salt solution containing 10 vol% FCS. Then, the cells were centrifuged at 300 g, 4 °C for 3 min. After removing the supernatant, 50 μL of secondary mAbs suspended in Hank's balanced salt solution containing 10 vol% FCS was added. The cells were incubated at 4 °C for 20 min, and washed with Hank's balanced salt solution containing 10 vol% FCS. Then, the cells were centrifuged at 300 g, 4 °C for 3 min. After removing the supernatant, the cells were suspended in an appropriate volume of Hank's balanced salt solution containing 10 vol% FCS. The concentration of the mAbs is provided in Supplementary Data 1. The cells were then analyzed on a FACSAria IIIu or FACSCanto II flow cytometer (BD Biosciences). Data were analyzed using FlowJo software (v10.8.1, BD Biosciences). ILC2s were defined as CD45$^+$lineage$^-$ (CD3ε, CD4, CD8α, CD11c, FcεRIα, NK1.1, CD19, TER119, F4/80, Ly-6G, and Ly-6C) Thy-1$^+$ST2$^+$KLRG1$^+$; ST2$^-$KLRG1$^-$ cells (ILC3-like cells) were defined as CD45$^+$lineage$^-$Thy-1$^+$ST2$^-$KLRG1$^-$; Siglec-F$^+$CD11c$^+$ alveolar macrophages were defined as CD45$^+$Gr-1$^-$F4/80$^+$Siglec-F$^+$CD11$^+$; Siglec-F$^{low}$CD11c$^+$ macrophages were defined as CD45$^+$Gr-1$^-$F4/80$^+$Siglec-F$^{low}$CD11$^+$; epithelial cells were defined as TER119$^-$CD31$^-$CD45$^-$EpCAM$^+$PDGFRα$^-$; and fibroblasts were defined as

TER119⁻CD31⁻CD45⁻EpCAM⁻PDGFRα⁺. The gating strategies are shown in Supplementary Fig. 5b, 5e, and 8a.

### Sorting ILC2s and fibroblasts, and epithelial cells

Cells were isolated from the lungs as described above. To simultaneously sort ILCs and fibroblasts, cells were stained with an appropriate concentration of purified mAbs against mouse CD16/CD32 and biotin-conjugated mAbs against lineage markers (CD3ε, CD4, CD8α, CD11c, FcεRIα, NK1.1, CD19, TER119, F4/80, and Gr-1) and against CD31 and EpCAM. The cells were washed and incubated with an appropriate concentration of streptavidin microbeads (#130-048-101, Miltenyi Biotec), following the manufacturer's instructions. Lineage⁻CD31⁻EpCAM⁻ cells were enriched using an autoMACS system (Miltenyi Biotec) and stained with fluorochrome-conjugated mAbs and fluorochrome-conjugated streptavidin. Specifics of the staining method can be found in the preceding section (Flow cytometry analysis). The concentration of the mAbs is provided in Supplementary Data 1. After staining, propidium iodide (0.5 µg·mL⁻¹) was added to detect the dead cells. CD45⁺lineage⁻Thy-1⁺ST2⁺KLRG1⁺ cells were identified as ILC2s, while CD45⁻lineage⁻CD31⁻EpCAM⁻PDGFRα⁺ were identified as fibroblasts. ILC2s and fibroblasts were then sorted using a FACSAria IIIu flow cytometer (BD Biosciences). The gating strategy is shown in Supplementary Fig. 9.

To simultaneously sort epithelial cells and fibroblasts, cells were stained with an appropriate concentration of purified mAbs against mouse CD16/CD32 (2 µg·mL⁻¹) and biotin-conjugated mAbs against TER119, CD31, and CD45 (0.5–2.5 µg·mL⁻¹). The cells were then washed and incubated with an appropriate concentration of streptavidin microbeads (Miltenyi Biotec), following the manufacturer's instructions. TER119⁻CD31⁻CD45⁻ cells were enriched using an autoMACS system (Miltenyi Biotec) and stained with fluorochrome-conjugated mAbs and fluorochrome-conjugated streptavidin. Specifics of the staining method can be found in the preceding section (Flow cytometry analysis). The concentration of the mAbs is provided in Supplementary Data 1. After staining, propidium iodide (0.5 µg·mL⁻¹) was added to detect the dead cells. TER119⁻CD31⁻CD45⁻EpCAM⁺PDGFRα⁻ cells were identified as epithelial cells, while TER119⁻CD31⁻CD45⁻EpCAM⁻PDGFRα⁺ cells were identified as fibroblasts. Both epithelial cells and fibroblasts were then sorted using a FACSAria IIIu flow cytometer (BD Biosciences). The gating strategy is shown in Supplementary Fig. 8c.

### scRNA-seq and related experimental procedures

Whole lung cells were collected from *Ifngr1⁻/⁻Rag2⁻/⁻* (7, 12, 16, 19, and 24 weeks old, *n* = 2 at each age) and control *Rag2⁻/⁻* (18 weeks old, *n* = 2) mice, as described above. Single-cell encapsulation and library construction were performed using a Chromium Controller with a Chromium Single-Cell 3' v2 Reagents Kit (10x Genomics, Pleasanton, CA, USA), following the manufacturer's instructions. Libraries were sequenced using a NovaSeq instrument (Illumina, San Diego, CA, USA). Gene counts were obtained by aligning reads to the mm10 genome using Cell Ranger software (v3.1.0, 10x Genomics). The initial quality control and downstream analyses were conducted using the Seurat package (v4.3.0)[41,42] of the R software (v4.1.3). All cells from different samples were integrated into one Seurat object and filtered based on unique feature counts (cells with 200–5,000 unique feature counts were treated as a single-cell and included) and mitochondrial counts (cells with >5% mitochondrial genes were treated as apoptotic cells and removed). In total, 59,332 cells were retained after filtering. The feature expression for each cell was then normalized to the total expression, multiplied by a scale factor (set to 10,000), and natural-log transformed using log1p. Next, a linear transformation was applied to the data: the expression of each gene was shifted and scaled, so that the mean expression across cells is 0, and the variance across cells is 1. Following that, PCA was performed on the scaled data, on the top 2000 genes with high standardized variance between cells. Standardized variance was calculated, calling Surat's FindVariableFeatures function (selection.method = "vst"). Following that, a K-nearest neighbor (KNN) graph on the euclidean distance in PCA space was constructed using the previously defined dimensionality of the dataset (first 25 principal components) as input, and graph-based clustering was performed using the Louvain algorithm[43]. Thereafter, non-linear dimensional reduction by the Uniform Manifold Approximation and Projection (UMAP) method was performed using the previously defined dimensionality of the dataset (first 25 principal components) as input. The cell types of each cluster were first determined by transferring cell type labels from a reference dataset[44] to our dataset, using Seurat's FindTransferAnchors function, which takes the previously defined dimensionality of the dataset (first 25 principal components) as input. After automatic cell type prediction, the cell type definition was further corrected manually with known cell type-specific markers. Cells identified as ILCs (3047 cells) were subjected to further PCA, sub-clustering (using first 25 principal components as input), and non-linear dimensional reduction using UMAP (using first 25 principal components as input).

### Downstream analysis of scRNA-seq data

We calculated the frequency of the cells of interest in either *Ptprc*-positive cells, *Ptprc*-negative cells, or ILCs, separately for each phase. To estimate the frequency changes over disease progression, we next calculated the log₂FC of the frequency in either the inflammatory or the fibrotic phase and compared it to that in the intact phase. Differentially expressed genes in each cluster were identified using the Seurat package. To define ILC scores, we used previously reported method[22]. Briefly, we calculated the arithmetic mean of the log-normalized feature counts of signature genes in ILC1s, ILC2s, or ILC3s. The log-normalized feature counts were calculated, using Surat's NormalizeData function: feature counts for each cell are divided by the total counts for that cell and multiplied by 10,000, which is then natural-log transformed using log1p. Genes reported by Wallrapp et al.[22] were used for the ILC signatures. The Seurat or ggplot2 package (v3.4.0) the R software was used for data visualization.

### ILC2s and fibroblasts co-culture

ILC2s and fibroblasts were sorted from the lungs of WT mice as described above. ILC2s (5000 cells/well) and fibroblasts (20,000 cells/well) were co-cultured in 96-well flat-bottom plates (Corning, Corning, NY, USA) coated with 0.1 vol% gelatin (Sigma-Aldrich) in complete RPMI-1640 culture medium (RPMI-1640 medium supplemented with 10 vol% heat-inactivated FCS, 100 U/ml penicillin-streptomycin, 10 mM HEPES buffer solution, 1 × MEM nonessential amino acids, 1 mM sodium pyruvate, 50 µM 2-mercaptoethanol and 50 µg/ml gentamycin sulfate)[45]. IL-2 (R&D Systems), -7 (Miltenyi Biotec), and -33 (R&D Systems) were added to the wells, at concentrations of 10 ng·mL⁻¹ each. On day 5, ILC2s were washed out, and fibroblasts were stained with Sirius Red.

### Sirius red staining

After washing with PBS, the fibroblasts co-cultured with ILC2s were fixed with 10 vol% formalin suspended in PBS. The cells were rinsed with PBS twice, followed by the addition of Picro-sirius Red Solution (0.1 wt% Sirius red [#365548-5 G, Sigma-Aldrich] in saturated aqueous solution of picric acid [Sigma-Aldrich]). After a 1-h incubation, the cells were washed twice with acidified water (0.5 vol% acetic acid [Nacalai Tesque] in distilled water)[46]. After staining, the cells were observed using a BZ-X700 microscope (Keyence, Osaka, Japan). Sirius Red was then eluted using 0.1 M NaOH (Wako, Tokyo, Japan) + methanol (Sigma-Aldrich) (1:1, vol/vol) (50 µL/well), and the absorption rate was measured at 540 nm (reference wavelength, 655 nm) using a microplate reader (iMark, Bio-Rad).

### *Il33* quantification using real-time quantitative PCR (qPCR)

Lungs were isolated immediately after sacrificing the mice. The lungs of 22-week-old mice were separated into normal and disease areas for analysis, whereas whole lungs of 7-week- and 14-week-old mice were collected for the analysis due to the absence of any lesions in them. The disease area was defined as a visibly distinct white region, as depicted in Fig. 1a. The lungs were then frozen in liquid nitrogen and broken into pieces using a crusher (SK-100; Tokken; Chiba, Japan). RNA was extracted using TRIzol Reagent (#15596026, Thermo Fisher Scientific) and complementary DNA was synthesized using a Super-Script III First-Strand Synthesis System for RT-PCR (#18080051, Thermo Fisher Scientific), according to the manufacturer's instructions. qPCR was performed using TB Green Premix Ex Taq (#RR420S, TaKaRa, Shiga, Japan) on an ABI 7500 Real-Time PCR System (Applied Biosystems, Waltham, MA, USA). Relative gene expression levels were calculated using the comparative threshold cycle method (ddCt)[47]. ddCt was calculated as

$$ddCt = (Ct_{control(reference\ gene)} - Ct_{sample(reference\ gene)})$$
$$- (Ct_{control(gene\ of\ interest)} - Ct_{sample(gene\ of\ interest)})$$

One sample in the 7-week-old group was used for normalization as a control.

The following primers were used in this study.
*Actb:*
(Forward) 5'-ACTATTGGCAACGAGCGGTTC-3'
(Reverse) 5'-GGATGCCACAGGATTCCATAC-3'
*Il33:*
(Forward) 5'-TCCAACTCCAAGATTTCCCCG-3'
(Reverse) 5'-CATGCAGTAGACATGGCAGAA-3'

The expression of *Il33* relative to *Actb* was normalized using one sample from 7-week-old mice as a control.

### IL-33 quantification using ELISA

Fibroblasts and epithelial cells were sorted from the lungs of *Ifngr1⁻/⁻Rag2⁻/⁻* mice as described above (30,000 cells/head in a 1.5 mL tube). After sorting, the collection tubes were centrifuged, and the resulting pellets were stored at −80 °C until analysis. Cells were then suspended in 100 μL of RIPA buffer (#ab156034, Abcam, Cambridge, UK) with a protease inhibitor tablet (#05892791001, Roche) and then sonicated to lyse the cells (SONIFIRE 250 A, Emerson, Kanagawa, Japan). Next, the tubes were centrifuged at $16,000 \times g$, 4 °C for 10 min, and the supernatant was collected for IL-33 quantification. IL-33 levels were determined by means of ELISA using a Quantikine Kit (#M3300, R&D Systems) following the manufacturer's instructions. The absorption rate was measured at 450 nm (reference wavelength, 570 nm) using a microplate reader (iMark, Bio-Rad).

### Masson's trichrome (MT) staining

The lung tissues were fixed for 24 h in 10 vol% formalin suspended in PBS and embedded in paraffin. The sections were prepared at a thickness of 3 μm. For MT staining, sections were stained with Wei-gert's iron hematoxylin, 0.75% Orange G solution, Masson's stain solution B, and an Aniline Blue solution (Muto Pure Chemicals, Tokyo, Japan), following the manufacturer's instructions. Sections were then observed under a BZ-X700 microscope (Keyence) or a SZX7 micro-scope (OLYMPUS, Tokyo, Japan).

### Analysis of Masson's trichrome-stained images

A quantitative analysis pipeline was established to automatically detect and quantify the ratio of the collagen-positive area to the entire area in the lung tissue section.

**Step 1: Pre-processing of the image.** To avoid undesired artifacts in image processing, the four edges of each image were first filled with the mean value of the white background of the image. The white background was estimated via extraction of pixels, with a smaller coefficient of variation among the three-color channels when compared to a threshold. A grayscale image was then created by averaging the three-color channels, after which the alveolar-wall regions that showed smaller pixel values than the background were segmented using a threshold. The resulting binary mask image that represented the alveolar-wall regions was used in the subsequent image-processing steps.

**Step 2: Estimation of the entire lung tissue region.** The alveolar-wall binary image (obtained in step 1) was dilated such that the fragmented alveolar-wall regions remain connected with each other. To remove unrelated artifacts from the image, connected contours with an area smaller than a threshold were discarded. The internal pixels of the remaining connected contours were filled, after which the resulting binary image was eroded to cancel the effect of the initial dilation at the edges of the contour. This produced a binary image that precisely covered the entire lung tissue section.

**Step 3: Estimation of the fibrotic regions.** In fibrotic regions, the alveolar wall is thickened by collagen and shows a dense structure wherein the normal alveolar structure is destroyed. To reflect this property in our image-processing algorithm, two-dimensional moving averaging was performed on the alveolar-wall binary image (obtained in step 1) with the aim of deriving the fibrotic regions in each image of the lung tissue section. The resulting image showed higher pixel values in densely packed fibrotic regions and lower pixel values in normal regions where the regular, sparse alveolar-wall structures were pre-served. Therefore, high-intensity regions were segmented by a threshold and used as the low-resolution binary estimate of the fibrotic region. The original color image was multiplied by the resulting low-resolution binary image and the high-resolution alveolar-wall binary image (obtained in step 1) to produce an image of the fibrotic regions.

**Step 4: Estimation of the collagen-positive regions.** To quantita-tively extract collagen-positive regions without human bias, principal component analysis (PCA) was performed in the color channel of the fibrotic region image (obtained in step 3). The second or third com-ponent of the spectral PCA was found to reflect the transitions of the stained color between blue (i.e., collagen-positive) and red (fibrin-positive or blood vessel). Therefore, the collagen-positive regions were defined via segmentation of the second or third component of the image produced via spectral PCA with a threshold. The order of the component and the polarity of the coefficient depended on the image; however, they were automatically detected and compensated for.

**Step 5: Calculation of the fibrosis score.** The fibrosis score was defined as the ratio of the area of the collagen-positive regions (obtained in step 4) to the area of the entire lung tissue region (obtained in step 2).

The code for this analysis is provided in the Code Availability section. The procedure of the analysis is described in Supplementary Fig. 2a. Image analysis was performed using MATLAB (R2023a).

### Statistical analysis of fibrosis scores

For Fig. 4b, an equal number of *Ifngr1⁻/⁻Rag2⁻/⁻* to those of *Ifngr1⁻/⁻Il2rg⁻/⁻Rag2⁻/⁻* mice ($n = 10$ mice) were randomly sampled three times from the pool of *Ifngr1⁻/⁻Rag2⁻/⁻* mice in the fibrosis stage ($n = 27$ mice, shown in Supplementary Fig. 2c) and the fibrosis scores were statistically tested using a two-tailed unpaired Student's *t*-test. Similar results were obtained in all three analyses and representative results are shown in Fig. 4b. The same strategy was used for both Figs. 5h and 6e. For Fig. 5h, an equal number of *Ifngr1⁻/⁻Rag2⁻/⁻* to those of *Ifngr1⁻/⁻Rorc⁻/⁻Rag2⁻/⁻* mice ($n = 7$ mice) were randomly sampled. For

Fig. 6e, an equal number of *Ifngr1-/-Rag2-/-* to those of *Ifngr1-/-Rag2-/-Il33gfp/gfp* mice (*n* = 8) were randomly sampled.

## Immunofluorescence staining

The tissues were embedded in an OCT compound (Sakura Finetek, Tokyo, Japan) and quickly frozen in liquid nitrogen. Sections were prepared at a thickness of 6 μm, immediately fixed in acetone-methanol (1:1, vol/vol) for 10 min, dried on slides for 1 h, and kept at −80 °C until staining. Sections were washed with 0.05 vol% Tween20 suspended in PBS (PBST) during staining. Endogenous biotin was first blocked for 2 h, at 37 °C, following the manufacturer's instructions (Endogenous Biotin-Blocking Kit, #E21390, Thermo Fisher Scientific). Subsequently, the sections were blocked with PBST containing 5 vol% FCS for 2 h, at 24 °C. Following this, the Fc receptors were blocked with an appropriate concentration of purified mAbs against mouse CD16/CD32, for 2 h, at 24 °C. The slides were then washed with PBST. Next, The slides were incubated with primary antibodies for 30 min, at 38.5 °C, and then washed with PBST. Subsequently, the slides were incubated with secondary antibodies for 30 min, at 38.5 °C, and then washed with PBST. The concentration of the mAbs is provided in Supplementary Data 1. Thereafter, nuclear staining was performed with 4′,6-diamidino-2-phenylindole (DAPI) for 15 min, at 24 °C, followed by a wash with PBST. Cover glass was then placed on the sections, and the sections were observed using a confocal microscope (TCS SP8, Leica Japan, Tokyo, Japan) or a BZ-X700 microscope (KEYENCE). The primary antibodies included mAbs specific for mouse GATA3 (#560078; L50-823, BD Biosciences), PDGFRα (#ab203491; EPR22059-270, Abcam), EpCAM (#118204; G8.8, BioLegend), and IL33 (#AF3626, R&D Systems). Secondary polyclonal antibodies against goat IgG (#A11055) and rabbit IgG (#A11010), and fluorochrome-conjugated streptavidin (#S11225) were purchased from Thermo Fisher Scientific.

## Analysis of immunofluorescence staining images

In the 21-week image, the area where the normal alveolar structure is preserved was defined as the normal area, and the area where the regular alveolar structure is disrupted was defined as the disease area. No distinction was made between areas for other samples because the 8-week-old samples had no disease areas and the 29-week-old samples had no normal areas.

Images were processed by channel. First, a threshold was set, and the image was binarized. Since the same cell may have intermittent regions when binarized, binarized images were dilated to connect the intermittent regions to form a single region, after which the resulting images were eroded to cancel out the effect of the initial dilation. Then objects smaller than a threshold were removed to exclude non-specifically stained small structures, and the number of remained objects (connected component) per image was counted. Fifty-one images of different ROIs (regions of interest) were analyzed. The code for this analysis is provided in the Code Availability section. The procedure of the analysis is described in Supplementary Fig. 8e. Image analysis was performed using MATLAB (R2022a).

## Participants and preparation of cell suspensions from human samples

Whole blood was obtained from 12 healthy adult volunteers (normal controls) from the Yokohama Minoru Clinic (Kanagawa, Japan) under a protocol approved by the Ethics Committee at the RIKEN Center for Integrative Medical Sciences (approval number: H29-12[5]), and Astellas Pharma Inc. (approval number: 000181). IPF whole blood was obtained from 20 individuals with IPF from Tokai University Hospital (Kanagawa, Japan) under a protocol approved by the Ethics Committee at the RIKEN Center for Integrative Medical Sciences (approval number: H29-12[5]), the Institutional Review Board for Clinical Research, Tokai University (approval number: 17R-157), and

Astellas Pharma Inc (approval number: 000181). All the healthy volunteers and patients with IPF provided written informed consent. All methods included in this study were conducted in accordance with relevant guidelines and regulations. IPF was diagnosed based on a multidisciplinary discussion by two pulmonologists and a radiologist using the patients' clinical history, physical examination, laboratory test results, and radiographic data from high-resolution computed tomography. Patients taking oral corticosteroids >10 mg prednisolone/day or those with concomitant allergic diseases such as asthma and eczema requiring pharmacological treatments were excluded. A description of the characteristics of the study population can be found in the Supplementary Data 2. Peripheral blood mononuclear cells were isolated from the whole blood using a BD Vacutainer CPT Mononuclear Cell Preparation Tube (#362753, BD Biosciences), suspended in CELLBANKER 1 plus (#11910, TaKaRa), and stored at −80 °C until analysis.

## Sorting of human ILC2s

Peripheral blood mononuclear cells were isolated from the whole blood, as described above. To sort ILC2s, cells were stained with an appropriate concentration of Zombie dyes (#423101, BioLegend), purified fluorochrome-conjugated mAbs against human fluorochrome-conjugated mAbs against lineage markers (CD3, CD4, CD14, CD16, CD19, and FcεRIα), and fluorochrome-conjugated mAbs against ILC2 markers (CD45, CD161, CRTH2, and CD127). The concentration of the mAbs is provided in Supplementary Data 1. Lineage−CD45+CD161+CRTH2+CD127+ cells were identified as ILC2s and sorted using a FACSAria III flow cytometer (BD Biosciences). Specifics of the staining method can be found in the preceding section (Flow cytometry analysis; Sorting ILC2s and fibroblasts, and epithelial cells). The gating strategies for the ILC2s are shown in Supplementary Fig. 10.

## RNA-seq of human ILC2s, experimental procedures

The ILC2s of 12 healthy controls and 20 individuals with IPF were sorted as described above. Library construction was performed using a NEBNext Single-Cell/Low Input RNA Library Prep Kit for Illumina (#E6420S, New England Biolabs Japan Inc., Tokyo, Japan), following the manufacturer's instructions. One sample from a patient with IPF was omitted before sequencing because of the low RNA quality. The libraries were sequenced using a HiSeq X instrument (Illumina). Quality control was performed using fastp (v0.21.0)[48]. Gene counts were obtained by aligning reads to the hg38 genome using HISAT2(v2.2.1)[49], followed by processing with samtools (v1.12)[50] and the Rsubread package (v2.4.3)[51] of the R software.

## RNA-seq of human ILC2s and downstream analysis

Normalization and downstream analyses were performed using the DESeq2 package (v1.30.1)[52] of the R software. Pre-filtering was performed to exclude genes with low or abnormally high expression levels (genes with mean expression levels below and above 2 SD, respectively). Filtered data were normalized using variance stabilizing transformation and subjected to PCA. Alternatively, filtered data were normalized and subjected to differential expression analysis, using the DESeq function of DESeq2, and then log fold changes were moderated using the lfcShrink function of DESeq2. Gene ontology enrichment analysis was performed with the highly expressed genes in ILC2s in the individuals with idiopathic PF compared to the healthy individuals ($\log_2$[fold change] > 5; $P < 10^{-25}$) using the clusterProfiler package (v3.14.3)[53] of the R software. The length of the genes was calculated using GenomicRanges (v1.42.0)[54] of the R software, and the transcripts per kilobase million (TPM) values were calculated. Z-scores of the TPM values of the fibrosis-related genes, which were selected based on the previous report[25], were visualized using the pheatmap package (v1.0.12) of the R software.

## Image processing

Representative images of MT staining, immunofluorescence staining, and Sirius Red staining were processed using Adobe Photoshop 2022 (Adobe, San Jose, CA, USA), and the brightness and contrast were adjusted. The color balance was adjusted for representative images of MT staining at low magnification and the immunofluorescence staining images shown in Supplementary Fig. 8d, left. The same process was applied to all images within the same figure.

## Statistical analysis

In all animal experiments, age- and sex-matched mice were randomly assigned to groups. No statistical methods were used to pre-determine the sample size. All experiments were performed using multiple biological replicates. The sample size was determined based on previous experience from studies by our group and those in the literature. The investigators were not blinded to sample identity and experimental conditions. All experiments, with the exceptions of Supplementary Fig. 1a, Fig. 2c, Supplementary Fig. 3d, Supplementary Fig. 5f, scRNA-seq, and bulk RNA-seq, were conducted multiple times and consistently yielded reproducible results, as described in the respective figure legends. Data are presented as the mean ± s.e.m. For statistical analysis, the following tests were used: one-way ANOVA with Tukey's multiple comparisons test, two-way ANOVA with Sidak's multiple comparisons test, one-way ANOVA with Dunnett's multiple comparisons test, two-tailed unpaired Student's $t$-test, two-tailed paired Student's $t$-test, two-sided Wilcoxon Rank Sum test with the Bonferroni method, two-sided Mann-Whitney test, and over representation analysis (ORA), which corresponds to a one-sided Fisher's exact test, with the Benjamini–Hochberg method. All statistical analyses were performed using Prism version 9 (GraphPad, Boston, MA, USA) or the R software. Data distribution was assumed to be normal, but this was not formally tested.

## Reporting summary

Further information on research design is available in the Nature Portfolio Reporting Summary linked to this article.

## Data availability

The scRNA-seq data of ILC2s from lungs of $Ifngr1^{-/-}Rag2^{-/-}$ mice generated in this study have been deposited in the NCBI GEO database under accession code GSE164220. The RNA-seq data of ILC2s from human serum have been deposited in the NCBI GEO database under accession code GSE194244. All other data are available in the article and its Supplementary files or from the corresponding author upon request. Source data are provided with this paper.

## Code availability

Codes for image analysis used in this study have been deposited in the Zendo open data repository under a following DOI [https://doi.org/10.5281/zenodo.10016650]. Other codes are available from authors upon reasonable request.

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

## Acknowledgements

We wish to thank Naho Hagiwara and Satsuki Tada for providing animal care and assistance with experiments; Hitomi Kodera, Tomoe Shitamichi, Yoshie Sasako, and Saori Miyachi for providing animal care; Naoto Fujioka for helping with the RNA-seq analysis; and Tsuyoshi Kiniwa, Tetsuro Kobayashi, and Takuma Misawa for their helpful discussions. We also thank Haruka Yabukami for helping us with the scRNA-seq library preparation; Masaru Ishii for the advice regarding micro-CT experiments; Yukihiro Horio, Takahisa Takihara, and Keito Enokida for human sample collection and evaluation of clinical data; Takashi Emori for assistance with experiments; Mako Numazaki for coordinating human sample experiments; Nobuyasu Endo for helping with RNA-seq analysis; Yoichiro Iwakura for providing *Ifng*⁻/⁻ mice; Susumu Nakae for providing *Il33*^(gfp/gfp) mice; Sidonia Fagarasan for providing *Rorc*^(gfp/gfp) mice. We also thank all laboratory members for their support and discussion. This work was supported by a JSPS Grant-in-Aid for JSPS Research Fellows (JP18J12457 to N.O.), a JSPS Grant-in-Aid for Scientific Research on Innovative Areas (JP18H05046 to Y.M.), a JSPS Grant-in-Aid for Scientific Research (JP18H05286 to K.M.), a JSPS Fund for the Promotion of Joint International Research (22K21354 to K.M.), a grant from the Takada Science Foundation (to K.M.), and a grant from Astellas (to K.M.).

## Author contributions

N.O. designed and performed the experiments, analyzed the data, and wrote the manuscript. Y.M. and S.K. suggested critical experiments and discussed the results. T.T., T.K. and A.M. assisted with the scRNA-seq analysis. M.M. and N.T. technically supported the daily experiments. M.T. performed the image analyses for deriving the fibrosis scores. F.S. and J.K. assisted with the CT imaging analysis. H.K., T.O., Y.S. and K.A. assisted with collecting human samples and provided helpful advice from a clinical perspective. J.M., Y.N. and Y.S. planned and performed the library preparation for RNA-seq analysis of human samples and assisted with the data analysis. K.M. planned and supervised the project and wrote the manuscript.

## Competing interests

K. Moro received a grant from Astellas Pharma, Inc. All other authors declare no competing interests.
