## [Peer Review File · Nature Communications]

Activation of ILC2s through constitutive IFN γ signaling reduction leads to spontaneous pulmonary fibrosisEditorial Note: This manuscript has been previously reviewed at another journal that is not operating a transparent peer review scheme. This document only contains reviewer comments and rebuttal letters for versions considered at *Nature Communications*.

REVIEWERS' COMMENTS

Reviewer #1 (Remarks to the Author):

This study is important for understanding the pathogenesis of human IPF by elucidating the role of ILC2/ILC3 in the disease utilizing a previously uncharacterized new mouse model. The authors have performed additional experiments or refined descriptions to ensure of accuracy, which have further supported their conclusions. Most previous concerns have been adequately addressed and the manuscript has been significantly improved.

Minor point :

In the analysis in figure 5d, it is interesting to find that cluster 1 and 5, which may both be ILC3s, have differential kinetics during fibrosis. It appears that cluster 5 is slightly enriched in inflammatory state and reduced at the fibrotic state, whereas cluster 1 is increased at both the inflammation and fibrotic stages. The authors use ILC3-like cells to define the Lin⁻Thy1⁺KLRG1⁻ST2⁻ cells, which theoretically is composed of both cluster 1 and 5 cells. What genes are differentially expressed by cluster 1 compared with cluster 5 (eg. Il17a, Il17f, Il22)? Is it likely that these cluster 1-signature genes contribute to ILC3-exacerbated fibrosis? Will this be mechanisms outside of the ILC3-supported ILC2 expansion in this fibrosis model? Some discussions on these possibilities would be insightful.

Reviewer #2 (Remarks to the Author):

I remain enthusiastic about the novelty and importance of this paper given (1) fibrosis model development with clinically relevant features; (2) accounting for ILCs in fibrosis; and now (3) strong new data showing association of ILC2 to fibrotic regions. These latter data (Fig 3J) should be quantified across at least three mice and plotted (#ILC2s/unit area of

fibrotic versus non fibrotic tissue). The IPF data at the end (peripheral blood ILC2 profiling) is an important correlate. I recognize the challenge of deleting IL33 in the fibroblast compartment, and this should be mentioned as a limitation/next step.

Reviewer #4 (Remarks to the Author):

In reviewing the manuscript, “Activation of ILC2s through constitutive IFN γ signaling reduction leads to spontaneous pulmonary fibrosis” by Otaki et al, I was asked to assess whether the comments of reviewer #3 had been adequately addressed by the revisions. Reviewer #3’s concerns primarily had to do with the lack of quantitative data and statistical analysis to support the authors’ conclusions regarding the cell types responsible for driving lung fibrosis in the *Ifngr1*^{-/-}*Rag2*^{-/-} mouse model. In response, the authors developed a fibrosis score, which relies on lung tissue sections stained with Masson’s trichrome stain and downstream image analysis to measure the percentage of the entire lung that is collagen positive. The pipeline for this fibrosis score is presented in Extended Data Figure 2 and described in detail in the methods. The use of this newly developed fibrosis score, quantification of data from key experiments, and inclusion of data from key controls have addressed the main concerns of reviewer #3 and resulted in a much improved manuscript.

Below, I comment specifically on the extent to which the authors’ responses address each of the reviewer’s comments.

Q1 - The reviewer commented on the lack of quantitative data and corresponding analysis to support a number of key conclusions in the paper. The authors have now included quantitative analysis of fibrosis progression using the newly described “fibrosis score” and have included quantification of total cells in the bronchoalveolar lavage fluid (BALF) as an assessment of inflammation.

Q2 - The reviewer commented that key controls were missing in the assessment of fibrosis progression in the *Ifngr1*^{-/-}*Rag2*^{-/-} mouse model. Specifically, the *Rag2*^{-/-} controls were not the same age as the *Ifngr1*^{-/-}*Rag2*^{-/-} mice. Because spontaneous fibrosis develops with age in this model, these are important controls to include. The authors now include quantitative

data from individual *Ifngr1^{-/-}Rag2^{-/-}* and *Rag2^{-/-}* mice, which demonstrate that fibrosis develops with age in *Ifngr1^{-/-}Rag2^{-/-}* mice and not in *Rag2^{-/-}* animals. The authors should also include a representative image of Masson's trichrome staining from a similarly aged *Rag2^{-/-}* mouse.

Q3 - The reviewer commented on the lack of information in the methods or figure legends on how data on collagen content was obtained and analyzed. The authors now present quantitative data with adequate description of methods and figure legends describing data in Extended Data Figure 2 and Figure 1, which address the reviewer's concerns.

Q4 - The reviewer commented that there was not sufficient data presented to strongly support the claim that in the *Ifngr1^{-/-}Rag2^{-/-}* fibrosis initially develops in subpleural regions of the lung. This is an interesting observation and an important area of future investigation. The authors have removed any conclusions from the results section and have appropriately moved comments on these observations to the discussion section.

Q5 - The reviewer commented that any claims on inflammation should include an assessment of the kinetics of parenchymal inflammation. In response, the authors present data on the kinetics of total cells in the BALF as a surrogate measure of inflammation in Figure 1c. These data are consistent with the development of inflammation preceding fibrosis.

Q6 - The reviewer again expressed concerns about the lack of quantification of key data to support the conclusions about the development of fibrosis. In response, the authors have presented quantitative data assessing Masson's trichrome staining for experiments involving *Ifngr1^{-/-}Rag2^{-/-}* and *Rag2^{-/-}* mice (Figure 1), *Ifngr1^{-/-}Rag2^{-/-}* and *Ifngr1^{-/-}Il2rg^{-/-}Rag2^{-/-}* (Figure 4), *Ifngr1^{-/-}Rag2^{-/-}* and *Ifngr1^{-/-}Rorc^{-/-}Rag2^{-/-}* (Figure 5), and *Ifngr1^{-/-}Rag2^{-/-}* and *Ifngr1^{-/-}Rag2^{-/-}Il33gfp/gfp* mice (Figure 6). As the authors acknowledge, inclusion of quantification of fibrosis and inflammation has greatly improved the manuscript and has provided much better characterization of the model and the factors and cell types that might be driving the development of fibrosis.

Q7 - The reviewer commented on the lack of age-matched controls for the physiological measurements of lung function in the fibrosis model. In response, the authors point to inclusion of fibrosis scores and assessment of total cells in the BALF for individual mice at different ages among *Ifngr1^{-/-}Rag2^{-/-}* and *Rag2^{-/-}* mice. However, the data on physiological lung function (SpO₂ or Cst) still do not include age-matched *Rag2^{-/-}* controls. The assumption is that the *Rag2^{-/-}* controls would have normal SpO₂ or Cst because they are refractory to the development of fibrosis, but this has not been shown. The authors have only partially addressed the reviewer's concerns.

Q8 - The reviewer commented on the lack of quantitative data in the experiments involving Dexamethasone treatment (Figure 2 d -f in the current submission). In response, the authors have included quantification of total cells in the BALF as a surrogate measure of inflammation. The authors have addressed the reviewer's concerns about quantification of data and statistical analysis to support claims. However, the experimental design and analysis still do not allow for strong conclusions about the effect of the timing of Dexamethasone treatment on reducing fibrosis in this model. The lung sections were analyzed at different ages for condition 1 (early DEX treatment) and condition 2 (late DEX treatment). Therefore, it is difficult to determine whether the lack of pathology observed in the DEX treated group in condition 1 is a result of a lack of development of fibrosis or a transient decrease in immune cell infiltration. In other words, are the DEX treated mice in condition 1 still protected from the development of fibrosis at 28 weeks? This is an important question to address because the authors suggest (lines 204 - 206) that early treatment with steroids during the inflammatory phase could improve disease outcomes for pulmonary fibrosis in humans. If these additional experiments cannot be completed, then I recommend removing these statements regarding prevention of disease progression in humans.

Point-by-point response to reviewers' comments:

Reviewer #1

This study is important for understanding the pathogenesis of human IPF by elucidating the role of ILC2/ILC3 in the disease utilizing a previously uncharacterized new mouse model. The authors have performed additional experiments or refined descriptions to ensure of accuracy, which have further supported their conclusions. Most previous concerns have been adequately addressed and the manuscript has been significantly improved.

Minor point : In the analysis in figure 5d, it is interesting to find that cluster 1 and 5, which may both be ILC3s, have differential kinetics during fibrosis. It appears that cluster 5 is slightly enriched in inflammatory state and reduced at the fibrotic state, whereas cluster 1 is increased at both the inflammation and fibrotic stages. The authors use ILC3-like cells to define the Lin-Thy1+KLRG1-ST2-cells, which theoretically is composed of both cluster 1 and 5 cells. What genes are differentially expressed by cluster 1 compared with cluster 5 (eg. Il17a, Il17f, Il22)? Is it likely that these cluster 1-signature genes contribute to ILC3-exacerbated fibrosis? Will this be mechanisms outside of the ILC3-supported ILC2 expansion in this fibrosis model? Some discussions on these possibilities would be insightful.

Response:

We sincerely appreciate the reviewer's interest in our manuscript and their encouraging comments. In response to the reviewer's question, we conducted differential expression analysis between two ILC3 populations, specifically Cluster 1 and 5 (The attached data, the upper panel). We observed that both clusters express *Ccr6* (The attached data, the lower left panel). Notably, Cluster 5 expresses *Cd4*, *Ccr7*, and *Tnfrsf4*, whereas Cluster 1 lacks *Cd4* expression (The attached data, the upper panel and the lower left panel). Consequently, Cluster 5 appears to represent a typical LTI-like ILC3, and Cluster 1 seems to align with the ex-NKp46⁺ ILC3 subset previously reported by Verrier *et al* (*J Immunol.* 2016 Jun 1;196(11)). Additionally, we examined cytokine expression within Cluster 1 and Cluster 5, revealing that Cluster 5 exhibited higher levels of *Il22* expression compared to Cluster 1, with no significant differences in *Il17a* and *Il17f* (The attached data, the upper panel and the lower right panel).

While we have the option to present this data as extended data, given that our primary focus lies elsewhere and the classification and function of ILC3s in the lungs remain largely unexplored, we would prefer to describe these results in the main text (Page 10-11, line 303-306) without including the actual data.

Reviewer #2

I remain enthusiastic about the novelty and importance of this paper given (1) fibrosis model development with clinically relevant features; (2) accounting for ILCs in fibrosis; and now (3) strong new data showing association of ILC2 to fibrotic regions. These latter data (Fig 3J) should be quantified across at least three mice and plotted (#ILC2s/unit area of fibrotic versus non fibrotic tissue). The IPF data at the end (peripheral blood ILC2 profiling) is an important correlate. I recognize the challenge of deleting IL33 in the fibroblast compartment, and this should be mentioned as a limitation/next step.

Response:

We sincerely appreciate the reviewer's interest in our manuscript and their encouraging comments. The number of ILC2s per unit area in normal and fibrotic areas of three different *Ifngr1^{-/-}Rag2^{-/-}* mice was calculated and added as Extended data 5f (page 9, line 261-263). Furthermore, the significance of generating conditional knockout mice to identify the source of disease-causing IL-33 production is described in Discussion for future studies (page 15, line 463-467).

Reviewer #4

*In reviewing the manuscript, "Activation of ILC2s through constitutive IFN γ signaling reduction leads to spontaneous pulmonary fibrosis" by Otaki et al, I was asked to assess whether the comments of reviewer #3 had been adequately addressed by the revisions. Reviewer #3's concerns primarily had to do with the lack of quantitative data and statistical analysis to support the authors' conclusions regarding the cell types responsible for driving lung fibrosis in the *Ifngr1^{-/-}Rag2^{-/-}* mouse model. In response, the authors developed a fibrosis score, which relies on lung tissue sections stained with Masson's trichrome stain and downstream image analysis to measure the percentage of the entire lung that is collagen positive. The pipeline for this fibrosis score is presented in Extended Data Figure 2 and described in*

detail in the methods. The use of this newly developed fibrosis score, quantification of data from key experiments, and inclusion of data from key controls have addressed the main concerns of reviewer #3 and resulted in a much improved manuscript.

Below, I comment specifically on the extent to which the authors' responses address each of the reviewer's comments.

Q1 - The reviewer commented on the lack of quantitative data and corresponding analysis to support a number of key conclusions in the paper. The authors have now included quantitative analysis of fibrosis progression using the newly described "fibrosis score" and have included quantification of total cells in the bronchoalveolar lavage fluid (BALF) as an assessment of inflammation.

Response: We appreciate this comment from the reviewer.

*Q2 - The reviewer commented that key controls were missing in the assessment of fibrosis progression in the *Ifngr1^{-/-}Rag2^{-/-}* mouse model. Specifically, the *Rag2^{-/-}* controls were not the same age as the *Ifngr1^{-/-}Rag2^{-/-}* mice. Because spontaneous fibrosis develops with age in this model, these are important controls to include. The authors now include quantitative data from individual *Ifngr1^{-/-}Rag2^{-/-}* and *Rag2^{-/-}* mice, which demonstrate that fibrosis develops with age in *Ifngr1^{-/-}Rag2^{-/-}* mice and not in *Rag2^{-/-}* animals. The authors should also include a representative image of Masson's trichrome staining from a similarly aged *Rag2^{-/-}* mouse.*

Response: The fibrosis scores of 19 *Rag2^{-/-}* mice shown in Figure 3a are data quantified from Masson's trichrome staining. Representative data for *Rag2^{-/-}* mice have already been shown in Figure 1f.

Q3 - The reviewer commented on the lack of information in the methods or figure legends on how data on collagen content was obtained and analyzed. The authors now present quantitative data with adequate description of methods and figure legends describing data in Extended Data Figure 2 and Figure 1, which address the reviewer's concerns.

*Q4 - The reviewer commented that there was not sufficient data presented to strongly support the claim that in the *Ifngr1^{-/-}Rag2^{-/-}* fibrosis initially develops in subpleural regions of the lung. This is an interesting observation and an important area of future investigation. The authors have removed any conclusions from the results section and have appropriately moved comments on these observations to the discussion section.*

Q5 - The reviewer commented that any claims on inflammation should include an assessment of the kinetics of parenchymal inflammation. In response, the authors present data on the kinetics of total cells in the BALF as a surrogate measure of inflammation in Figure 1c. These data are consistent with the development of inflammation preceding fibrosis.

*Q6 - The reviewer again expressed concerns about the lack of quantification of key data to support the conclusions about the development of fibrosis. In response, the authors have presented quantitative data assessing Masson's trichrome staining for experiments involving *Ifngr1^{-/-}Rag2^{-/-}* and *Rag2^{-/-}* mice (Figure 1), *Ifngr1^{-/-}Rag2^{-/-}* and *Ifngr1^{-/-}Il2rg^{-/-}Rag2^{-/-}* (Figure 4), *Ifngr1^{-/-}Rag2^{-/-}* and *Ifngr1^{-/-}Rorc^{-/-}Rag2^{-/-}* (Figure 5), and *Ifngr1^{-/-}Rag2^{-/-}* and *Ifngr1^{-/-}Rag2^{-/-}Il33gfp/gfp* mice (Figure 6). As the authors acknowledge, inclusion of quantification of fibrosis and inflammation has greatly improved the manuscript and has provided much better characterization of the model and the factors and cell types that might be driving the development of fibrosis.*

Responses to Q3-Q6: We appreciate these comments from the reviewer.

*Q7 - The reviewer commented on the lack of age-matched controls for the physiological measurements of lung function in the fibrosis model. In response, the authors point to inclusion of fibrosis scores and assessment of total cells in the BALF for individual mice at different ages among *Ifngr1^{-/-}Rag2^{-/-}* and *Rag2^{-/-}* mice. However, the data on physiological lung function (SpO₂ or Cst) still do not include age-matched *Rag2^{-/-}* controls. The assumption is that the *Rag2^{-/-}* controls would have normal SpO₂ or Cst*

because they are refractory to the development of fibrosis, but this has not been shown. The authors have only partially addressed the reviewer's concerns.

Response:

We have measured SpO₂ levels in aged *Rag2*^{-/-} mice and confirmed that there is no difference in SpO₂ levels in young and old *Rag2*^{-/-} mice. Unfortunately, conducting C_{st} experiments was not feasible due to the initial experiments being conducted with a demo machine borrowed from the company. It is important to note that both SpO₂ and C_{st} data are typically correlated as they both depend on lung function. Therefore, we believe that the SpO₂ data alone provides sufficient information to address the reviewer's concerns. We have incorporated this data into Extended Data Figure 3d (page 7, line 187-188).

Q8 - The reviewer commented on the lack of quantitative data in the experiments involving Dexamethasone treatment (Figure 2 d -f in the current submission). In response, the authors have included quantification of total cells in the BALF as a surrogate measure of inflammation. The authors have addressed the reviewer's concerns about quantification of data and statistical analysis to support claims. However, the experimental design and analysis still do not allow for strong conclusions about the effect of the timing of Dexamethasone treatment on reducing fibrosis in this model. The lung sections were analyzed at different ages for condition 1 (early DEX treatment) and condition 2 (late DEX treatment). Therefore, it is difficult to determine whether the lack of pathology observed in the DEX treated group in condition 1 is a result of a lack of development of fibrosis or a transient decrease in immune cell infiltration. In other words, are the DEX treated mice in condition 1 still protected from the development of fibrosis at 28 weeks? This is an important question to address because the authors suggest (lines 204 - 206) that early treatment with steroids during the inflammatory phase could improve disease outcomes for pulmonary fibrosis in humans. If these additional experiments cannot be completed, then I recommend removing these statements regarding prevention of disease progression in humans.

Response: We understand the reviewer's concern and have removed the text on page 8, line 205-207.